# Mating and male pheromone kill *Caenorhabditis* males through distinct mechanisms

**Cheng Shi, Alexi M Runnels, Coleen T Murphy***

Department of Molecular Biology and LSI Genomics, Princeton University, Princeton, United States

**Abstract** Differences in longevity between sexes is a mysterious yet general phenomenon across great evolutionary distances. To test the roles of responses to environmental cues and sexual behaviors in longevity regulation, we examined *Caenorhabditis* male lifespan under solitary, grouped, and mated conditions. We find that neurons and the germline are required for male pheromone-dependent male death. Hermaphrodites with a masculinized nervous system secrete male pheromone and are susceptible to male pheromone killing. Male pheromone-mediated killing is unique to androdioecious *Caenorhabditis*, and may reduce the number of males in hermaphroditic populations; neither males nor females of gonochoristic species are susceptible to male pheromone killing. By contrast, mating-induced death, which is characterized by germline-dependent shrinking, glycogen loss, and ectopic vitellogenin expression, utilizes distinct molecular pathways and is shared between the sexes and across species. The study of sex- and species-specific regulation of aging reveals deeply conserved mechanisms of longevity and population structure regulation.

**\*For correspondence:** ctmurphy@princeton.edu

**Competing interests:** The authors declare that no competing interests exist.

## Introduction

Males and females differ in many aspects of biology, including longevity. Sex differences in lifespan are common in animals across great evolutionary distances (*Austad and Fischer, 2016*). For example, women live longer than men in almost every country (*WHO, 2016*). Moreover, interventions in longevity also display sex-specific patterns in mice (*Austad and Fischer, 2016*). However, the underlying mechanisms of sex differences in longevity and responses to aging interventions, and the degree of evolutionary conservation of these mechanisms, are still largely unknown.

Interactions between the sexes influence an individual's longevity (*Fowler and Partridge, 1989*; *Gems and Riddle, 1996*; *Partridge and Farquhar, 1981*; *Van Voorhies, 1992*). Sex-specific longevity patterns also exist in *Caenorhabditis elegans*: males live significantly shorter when maintained in groups, whereas the longevity of hermaphrodites is not influenced by population density (*Gems and Riddle, 2000*). What causes this sex-specific longevity pattern remains mysterious. Although worm studies have contributed significantly to our general understanding of longevity mechanisms, how *C. elegans* male lifespan is regulated is poorly understood, because nearly all lifespan experiments are performed using only hermaphrodites. Analysis of *Caenorhabditis* males' longevity not only allows us to test whether known longevity mechanisms are conserved between the sexes, but also provides an opportunity to reveal novel longevity mechanisms. The lifespan of *Caenorhabditis* females is shortened after mating through receipt of male sperm and seminal fluid (*Shi and Murphy, 2014*), and separately by exposure to male pheromone (*Maures et al., 2014*). However, previous studies reported contradictory results on the influence of mating on male lifespan (*Gems and Riddle, 1996*;

**eLife digest** In many animals, different sexes have different life expectancies. This holds true for a roundworm species called *Caenorhabditis elegans* that has commonly been used to study aging and lifespan. Unlike some related *Caenorhabditis* roundworm species (which consist of male and female worms), *C. elegans* worms are predominantly hermaphrodites and reproduce by self-fertilization. *C. elegans* males are normally rare. However, under stressful conditions the number of males increases to reduce inbreeding and so help the worm population to adapt to the environment.

Investigations into the factors that affect the lifespan of *C. elegans* have mostly studied hermaphrodites. For example, one recent study showed that mating shortens the lifespan of hermaphrodites. Another study showed that pheromones – hormones that change the behavior of other worms – also shorten hermaphrodite lifespan. The male pheromone is produced by males and sensed by both males and hermaphrodites. But does mating and male pheromone affect the lifespan of male roundworms?

Shi et al. have now studied *Caenorhabditis* worms of different species and sexes to investigate how sexual behaviors and male pheromone regulate the lifespan of male roundworms. The results of the experiments revealed two distinct mechanisms of male death. Firstly, mating caused the males of many different *Caenorhabditis* species to shrink and die, and also killed females and hermaphrodites. Secondly, the males of hermaphroditic species – and only these males – could also be killed by male pheromone.

The results suggest that death from mating may be an unavoidable cost of reproducing that is seen across all sexes and species of roundworm. In contrast, death by male pheromone may be a way of culling the male population in hermaphroditic species, for example, after stressful conditions have caused a sudden increase in the number of male worms.

Further work is now needed to investigate the finer details of the mechanisms by which mating and male pheromone cause death. Ultimately, this work in *Caenorhabditis* could be extended to help us to understand how other animals regulate their lifespan and maintain an optimum ratio of the sexes.

*Van Voorhies, 1992*). Thus, whether and how male lifespan is affected by prolonged exposure and interactions with females, as well as the effect of pheromone on male lifespan, is unknown.

In their natural environments, animals must not only find food, but also avoid competitors, identify appropriate partners, and mate; the effects of these behaviors on lifespan are not well understood. Some of these behaviors are mediated by ascaroside-based pheromones (*Ludewig and Schroeder, 2013*). *Caenorhabditis* females secrete pheromones that attract males (*Chasnov et al., 2007*), while *C. elegans* hermaphrodites modify their pheromone profile according to their sperm status, becoming more attractive to males once they have used up their own sperm (*Garcia et al., 2007*; *Kleemann and Basolo, 2007*; *Morsci et al., 2011*). Male ascaroside pheromones can directly affect the reproductive system of hermaphrodites, aiding recovery from heat stress and delaying the loss of hermaphrodite germline stem cells (*Aprison and Ruvinsky, 2015, 2016*).

The diversity of *Caenorhabditis* species allows us to evaluate male lifespan regulation from an evolutionary perspective. The *Caenorhabditis* genus consists of both androdioecious (male and hermaphrodite) and gonochoristic (male and female) species. In androdioecious species such as *C. elegans*, the population is dominated by hermaphrodites, which reproduce by self-fertilization. Males are usually very rare (less than 0.2% for the standard lab strain N2) and are produced through spontaneous X chromosome nondisjunction (*Chasnov and Chow, 2002*; *Hodgkin, 1983*). Under stressful conditions, more oocytes undergo chromosome non-disjunction; thus androdioecious species may periodically experience male population explosions (*Chasnov and Chow, 2002*; *Hodgkin, 1983*). The existence of males in androdioecious species reduces inbreeding and facilitates adaptation to changing environments (*Anderson et al., 2010*). By contrast, gonochoristic species such as *C. remanei* (50% male, 50% female) require mating for reproduction. How males in androdioecious and gonochoristic species cope with these different mating situations remains poorly understood. Moreover,

the utility of killing females by exposure to male pheromone in gonochoristic populations (*Maures et al., 2014*) is not obvious.

Here we have found that male-specific population density-dependent death in *C. elegans* is due to the perception of male pheromone as a toxin; that is, while male pheromone itself is not a general poison to all worms, its perception by *C. elegans* males leads to death and to male-specific reproductive decline. *C. elegans* hermaphrodites, while still susceptible, are less sensitive to this toxic aspect of male pheromone. Masculinization of the hermaphrodite nervous system not only increases their sensitivity to male pheromone, but also is sufficient to induce male density-dependent death in these hermaphrodites, suggesting that neurons are key for the male pheromone killing mechanism. We found that the germline of the recipient male is also required for male pheromone-mediated death. This phenomenon is also present in two other independently-evolved androdioecious *Caenorhabditis* species, suggesting a role for male pheromone killing of males in otherwise hermaphroditic species; by contrast, neither males nor females of three gonochoristic *Caenorhabditis* species succumb to male pheromone killing. Mating-dependent death and germline-dependent shrinking, by contrast, are shared between all sexes and *Caenorhabditis* species, suggesting deep conservation. Our work highlights the importance of understanding the shared vs. sex- and species-specific mechanisms that regulate lifespan.

## Results

### Male-specific density-dependent lifespan decrease is largely due to male pheromone

When *C. elegans* males are housed together, they live shorter than solitary males, and the death rate increases with the number of males in a dose-dependent manner (*Gems and Riddle, 2000*). *C. elegans* male lifespan is very sensitive to male density: just two males together significantly reduced each individual's lifespan, and grouping eight males decreases lifespan by more than 35%, whereas the lifespan of *C. elegans* hermaphrodites is not affected by population density (*Figure 1A,B*). It was shown previously that *C. elegans* hermaphrodites can be killed by pheromone secreted by grouped *C. elegans* males (*Maures et al., 2014*). We wondered whether the population density-dependent lifespan decrease of grouped *C. elegans* males is also due to male pheromone toxicity. To study the role of pheromone, we tested the survival of grouped *daf-22* (ascaroside pheromone-production deficient [*Golden and Riddle, 1985*]) males. Eight grouped *daf-22(m130)* males lived almost as long as solitary wild-type males, suggesting that male pheromone kills grouped wild-type males (*Figure 1C*; *Figure 1—figure supplement 1A*).

### Males are more sensitive to male pheromone toxicity than are hermaphrodites

The fact that adding just one other male significantly decreases male lifespan (*Figure 1A*) suggests that *C. elegans* males are very sensitive to male pheromone. To compare the sensitivity to male pheromone between the sexes, we decreased the number of males from 30–150 as was previously used in male-conditioned plates (*Maures et al., 2014*) to only eight per plate (*Figure 1—figure supplement 1B*), and examined the lifespans of *daf-22* males and hermaphrodites. This low dosage of male pheromone had no effect on hermaphrodites, but significantly reduced *C. elegans* male lifespan (*Figure 1D,E*), suggesting that males are more sensitive than hermaphrodites to male pheromone toxicity. In fact, we found that exposure to pheromone secreted by just one male was sufficient to significantly reduce the lifespan of *C. elegans* males (*Figure 1F*).

### Male pheromone toxicity requires the germline

To identify the transcriptional effects of male pheromone treatment on males, we performed expression analysis of *daf-22* (pheromone-less) males exposed to male pheromone for six days of adulthood. To avoid any germline-dependent effects of male interactions, we added the DNA replication inhibitor 5-fluorodeoxyruridine (FUdR) (*Figure 1G*) to the plates, which inhibits germline proliferation (*Shi and Murphy, 2014*). (Adult treatment with the DNA replication inhibitor 5-fluorodeoxyruridine (FUdR) has little effect on lifespan and meiosis at low dosage (50 μM [*Luo et al., 2009*]), but rapidly blocks germline proliferation in mated hermaphrodites (*Shi and Murphy, 2014*); FUdR and other

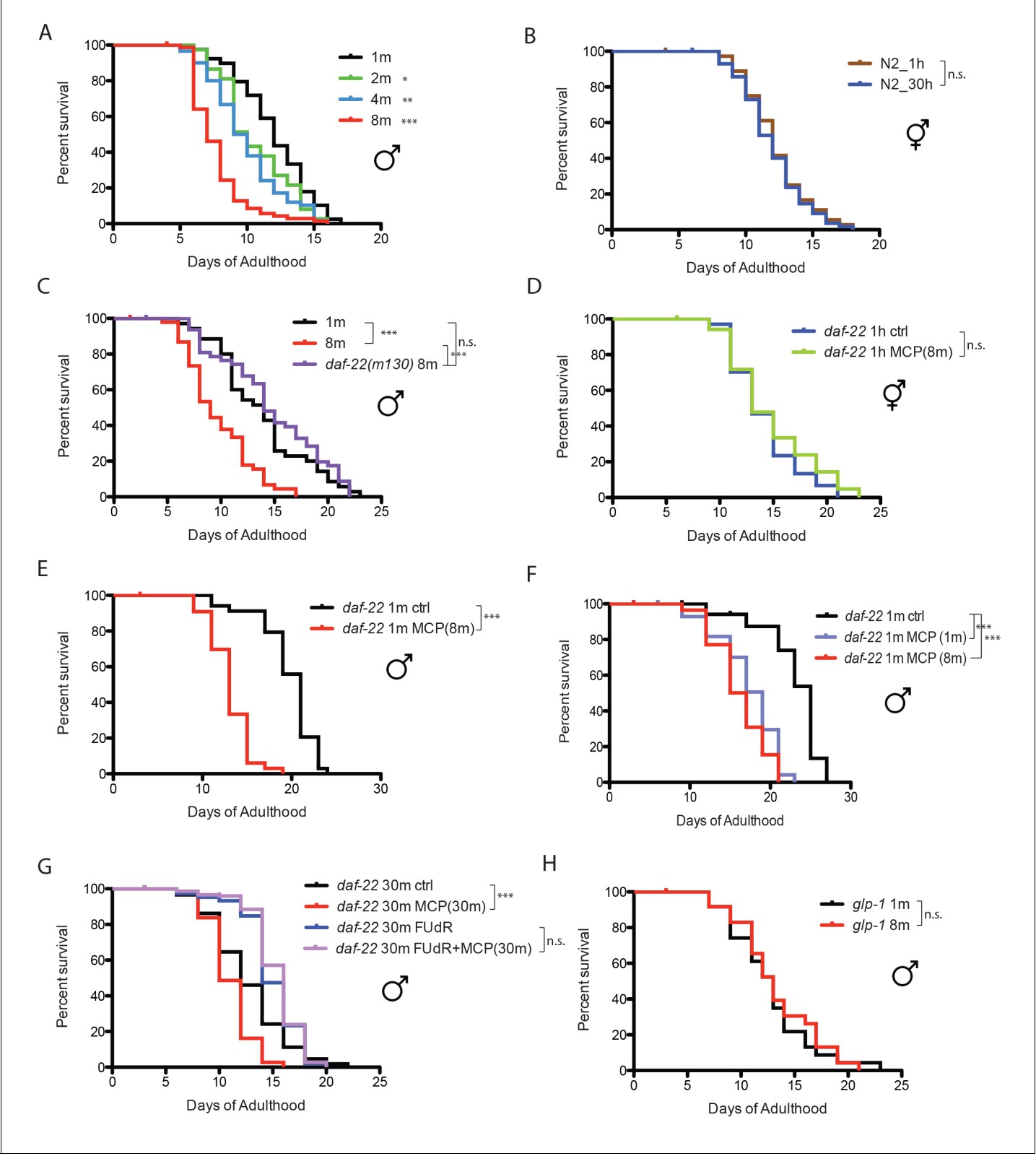

**Figure 1.** Male pheromone leads to early death in *C. elegans* males. (**A**) Lifespans of grouped *fog-2(q71)* males. We used *fog-2(q71)* mutants, as *fog-2* males are equivalent to wild-type (N2) males (*Schedl and Kimble, 1988*). Solitary males: 12.0 ± 0.4 days, n = 40; two males: 10.6 ± 0.4 days, n = 40, p=0.0397; four males: 9.9 ± 0.4 days, n = 60, p=0.0012; eight males: 7.7 ± 0.2 days, n = 80, p<0.0001. For all the lifespan assays performed in this study, Kaplan-Meier analysis with log-rank (Mantel-Cox) test was used to determine statistical significance. All the lifespan results are included in

*Figure 1 continued on next page*

*Figure 1 continued*

**Supplementary file 1.** (B) Lifespans of solitary N2 hermaphrodites: 12.3 ± 0.4 days, n = 40; grouped N2 hermaphrodites: 12.0 ± 0.3 days, n = 58, p=0.6436. (C) Grouped *daf-22(m130)* males have a similar lifespan to solitary wild-type *fog-2* males. (*daf-22(m130)* mutants are ascaroside pheromone-production deficient.) Solitary *fog-2* males: 13.8 ± 0.7 days, n = 35; eight *fog-2* males: 9.8 ± 0.5 days, n = 48, p<0.0001; eight *daf-22(m130)* males: 14.7 ± 0.7 days, n = 48, p=0.4039. Details about male-conditioned plates lifespan assays are included in Materials and methods and *Figure 1—figure supplement 1B*. (D) Lifespans of solitary *C. elegans daf-22(m130)* hermaphrodites on plates conditioned by eight *fog-2* males. Solitary *daf-22* hermaphrodites on control plates: 14.2 ± 0.6 days, n = 35; solitary *daf-22* hermaphrodites on male-conditioned plates: 14.8 ± 0.8 days, n = 35, p=0.4356. (E) Lifespans of solitary *C. elegans daf-22(m130)* males on plates conditioned by eight *fog-2* males. *daf-22(m130)* mutants are ascaroside pheromone-production deficient. Therefore, the effect of male pheromone is due to the male pheromone secreted by wild-type males when conditioning the plates. Solitary *daf-22* males on control plates: 19.7 ± 0.5 days, n = 34; solitary *daf-22* males on male-conditioned plates: 13.1 ± 0.4 days, n = 33, p<0.0001. (F) *daf-22(m130)* male lifespans on plates conditioned by wild-type *fog-2* males. Solitary *daf-22(m130)*: 23.0 ± 0.9 days, n = 30; *daf-22(m130)* on plates conditioned by one *fog-2* male: 17.3 ± 0.7 days, n = 29, p<0.0001; *daf-22(m130)* on plates conditioned by eight *fog-2* male: 16.1 ± 0.6 days, n = 30, p<0.0001. (G) Male pheromone-induced shorter lifespan of grouped *daf-22(m130)* males is inhibited in the presence of 50 µM FUdR. Grouped *daf-22* males on NGM: 12.7 ± 0.3 days, n = 150; grouped *daf-22* males on male-conditioned plates (MCP): 11.0 ± 0.2 days, n = 150, p<0.0001; grouped *daf-22* males on NGM with FUdR: 14.9 ± 0.2 days, n = 150, grouped *daf-22* males on MCP with FUdR: 15.3 ± 0.2 days, n = 150 p=0.2964. (H) Lifespans of grouped *glp-1(e2141)* males. Solitary males: 12.7 ± 0.8 days, n = 44; eight males: 13.3 ± 0.8 days, n = 56, p=0.699.
The following figure supplement is available for figure 1:

**Figure supplement 1.** Male pheromone-mediated toxicity requires germline.

germline-blocking approaches are commonly used in expression analyses to avoid confounds [*Reinke et al., 2000*; *Shaw et al., 2007*; *Maures et al., 2014*]. To our surprise, this comparison revealed no differences in gene expression (*Figure 1—figure supplement 1C*), even at a high % False Discovery Rate (FDR), suggesting that blocking male germline proliferation prevents male pheromone's effects on gene expression. Indeed, we found that there was no lifespan difference when *daf-22(m130)* males were subjected to a high level of exogenous wild-type male pheromone (30 males per plate for conditioning) in the presence of FUdR, mimicking the microarray conditions (*Figure 1G*). Furthermore, no population density-dependent lifespan decrease was observed when germline-deficient *glp-1(e2141)* males were grouped (*Figure 1H*). Similarly, exposing grouped *glp-1* hermaphrodites to very high levels of male pheromone (60 wild-type males per plate for conditioning) also failed to shorten lifespan or to induce significant transcriptional changes (*Figure 1—figure supplement 1D,E*). Therefore, our results suggest that germline activity in the recipient is required for male pheromone-mediated death in both sexes.

Interestingly, we also found that the loss of the germline itself also affects the production of male pheromone: males on plates conditioned by germline-less *glp-1* males lived longer than those on plates conditioned by wild-type males (*Figure 1—figure supplement 1F*), suggesting that there is communication between the status of germline and the production of male pheromone. However, males on plates conditioned by germline-less males still live shorter than the controls, suggesting that the germline cannot be the site of pheromone production, but rather may modulate pheromone levels or quality.

## Neuronal masculinization of hermaphrodites is sufficient to induce male-like pheromone production and sensitivity in hermaphrodites

The worm perceives environmental cues through its nervous system (*Bargmann, 2006*). To test whether the nervous system influences worms' sensitivity to male pheromone, we utilized a strain of *C. elegans* hermaphrodite in which the neurons have been masculinized (EG4389: *him-5(e1490) V; lin-15(n765ts)X; oxEx860[P(rab-3)::fem-3(wt)::mCherry(worm)::unc-54, pkd-2::gfp(S65C), lin-15(+)]*, a gift from the Jorgensen Lab [*White et al., 2007*]). Solitary neuronally-masculinized hermaphrodites died earlier when exposed to a low level of male pheromone (eight males per plate for conditioning; *Figure 2A*), whereas normal hermaphrodites were insensitive to male pheromone at this concentration (*Figure 1D*), suggesting that sex-specific neuronal properties are responsible for male and hermaphrodite's different sensitivities to male pheromone's toxicity.

Neuronal masculinization also changed the composition of pheromone secreted by the hermaphrodites: *C. elegans* males are normally attracted to pheromones secreted by old (self sperm-depleted) hermaphrodites (*Morsci et al., 2011*; *Leighton et al., 2014*; *Kleemann and Basolo,*

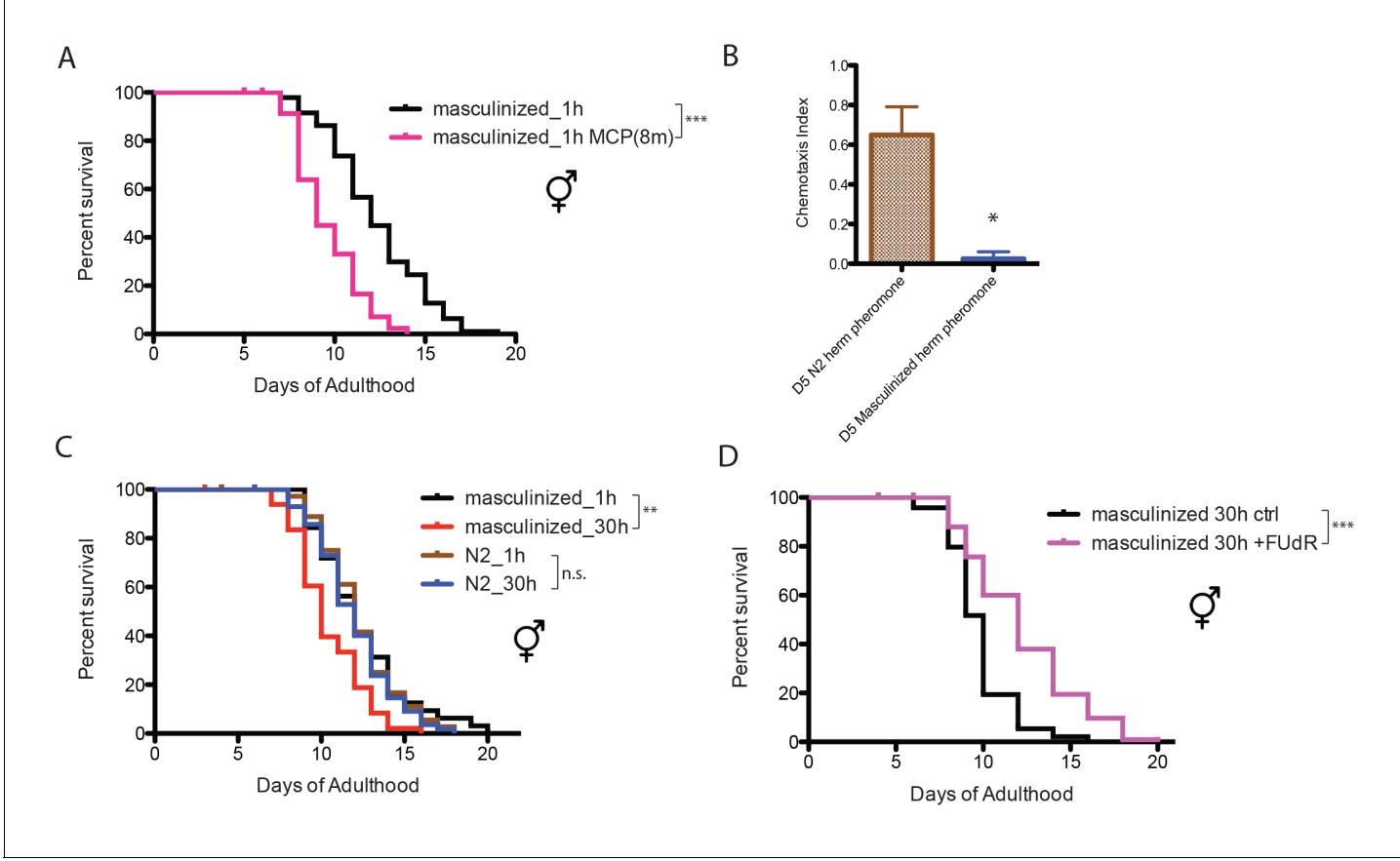

**Figure 2.** Neuronal masculinization of *C. elegans* hermaphrodites. (**A**) Neuronal masculinization of *C. elegans* hermaphrodites increases their sensitivity to male pheromone toxicity. Lifespans of solitary masculinized hermaphrodites: 12.3 ± 0.3 days, n = 96; solitary masculinized hermaphrodites on plates conditioned by eight males: 9.6 ± 0.3 days, n = 56, p<0.0001. (**B**) Supernatant solutions from Day 5 *C. elegans* hermaphrodites and masculinized hermaphrodites and Day one *fog-2* males were used to do the chemotaxis assay. See Materials and methods for detailed description. Chemotaxis Index (C.I.) of wild-type hermaphrodites' supernatant: 0.65 ± 0.10, C.I. of masculinized hermaphrodites' supernatant: 0.03 ± 0.03; p=0.0261, unpaired t-test. (**C**) Neuronal masculinization is sufficient to induce male-like population-density-dependent lifespan decrease in hermaphrodites. Lifespans of solitary N2 hermaphrodites: 12.3 ± 0.4 days, n = 40; grouped N2 hermaphrodites (about 30 worms per 35 mm plate): 12.0 ± 0.3 days, n = 58, p=0.6436. Solitary masculinized hermaphrodites: 12.4 ± 0.5 days, n = 40; grouped masculinized hermaphrodites (30 per plate): 10.4 ± 0.3 days, n = 60, p=0.0015. (**D**) FUdR rescue the lifespans of grouped masculinized hermaphrodites. Lifespans of grouped masculinized hermaphrodites: 9.8 ± 0.2 days, n = 111, grouped masculinized hermaphrodites (30 per plate) in the presence of 50 µM FUdR: 12.2 ± 0.3 days, n = 119, p<0.0001.

*2007*), but males were not attracted to pheromones secreted by aged neuronally-masculinized hermaphrodites (*Figure 2B*). More surprisingly, when these masculinized hermaphrodites were grouped, they lived shorter than the solitary controls (*Figure 2C*), suggesting that neuronal masculinization of the hermaphrodites is sufficient to induce the production of male-like pheromone in these hermaphrodites, and that neurons are key for male pheromone-mediated death. Inhibiting germline proliferation in these grouped masculinized hermaphrodites by FUdR rescued lifespan (*Figure 2D*), which again supports the conclusion that the germline, whether male or female, is required for male pheromone-mediated killing.

## *C. elegans* males shrink and die after mating

We previously found that mating greatly shortens hermaphrodite lifespan (*Shi and Murphy, 2014*), but the effect of mating on male lifespan is not yet known. To distinguish mating effects from the toxic effect of male pheromone, we measured the lifespans of solitary males and single males paired with a single hermaphrodite for 6 days from Day 1 to Day 6 of adulthood. (We used *fog-2(q71)* mutants, as *fog-2* males are equivalent to wild-type (N2) males, while *fog-2* hermaphrodites are self-

spermless (*Schedl and Kimble, 1988*), enabling identification of successful mating.) Mating decreased male lifespan ~35% compared with unmated solitary males (*Figure 3A*, *Supplementary file 1*), similar to the decrease in hermaphrodite lifespan caused by mating (*Shi and Murphy, 2014*). Males die faster when paired with a hermaphrodite for a longer period: mating with a hermaphrodite for one day did not significantly affect the lifespan of the male, while 2–3 days' mating shortened male lifespan by 15%, 4–5 days' mating reduced their lifespan by 25%, and mating for 6 days reduced lifespan more than 35% (*Figure 3B*). By contrast, the number of hermaphrodites paired with the single male during mating had little effect (*Figure 3C*). The time at which mating occurs within the reproductive period is also not critical for males' post-mating lifespan decrease; given the same mating duration, males mated with hermaphrodites for the first three days of adulthood had a similar lifespan decrease as those mated with hermaphrodites during days 6–8 of adulthood (*Figure 3D*).

As we previously observed in hermaphrodites (*Shi and Murphy, 2014*), males shrank after 6 days of mating; by Day 7, the mated males were 10% smaller than the unmated solitary males control (*Figure 3E,F*, *Supplementary file 2*). No such shrinking was apparent when males were treated with male pheromone (*Figure 3G*), suggesting that mating and male pheromone act through different pathways.

## Male post-mating shrinking and death depend on the germline

We wondered whether pheromone is required for mating-induced death in males; however, wild-type males still died early post-mating when paired with a *daf-22* hermaphrodite for 6 days (*Figure 4A*). Likewise, *daf-22* mutant males lived shorter after 6 days' mating with *fog-2* hermaphrodites (*Figure 4B*), indicating that the post-mating lifespan decrease in our single-worm pair lifespan assay is due to mating itself rather than to the presence of pheromone from either sex.

Elevated germline proliferation is one of the major causes of hermaphrodites' early death after mating (*Shi and Murphy, 2014*). This killing mechanism appears to be conserved in males: when treated with 50 µM FUdR during the three-day mating period, males no longer died earlier (*Figure 4C*). FUdR treatment also eliminated male post-mating lifespan decrease in our 6 day mating regime (*Figure 4—figure supplement 1A*). The absence of the germline also prevented mating-induced shrinking: germline-less *glp-1(e2141)* males experienced neither shrinking nor lifespan decrease after mating (*Figure 4D,E*, *Figure 4—figure supplement 1B*). These results suggest that germline-mediated post-mating death and shrinking is conserved between *C. elegans* sexes.

## Ectopic expression of vitellogenin contributes to male post-mating death

To identify molecular mechanisms that contribute to post-mating death in males, we performed genome-wide transcriptional analyses of mated and unmated males: we paired a single male with a hermaphrodite for 3.5 days of mating (150 pairs per biological replicate), then picked the males individually from the hermaphrodites on Day four for microarray analysis (*Figure 4—figure supplement 2A*). As a control, we collected the same number of age-matched solitary males. Only 14 genes were significantly up-regulated and 41 were significantly down-regulated (FDR = 0%; *Figure 4F*; *Supplementary file 3*). Genes whose expression decreased in mated males include extracellular proteins (*scl-11*, *scl-12*, *zig-4*) and predicted lipase-related hydrolases (*lips-11*, *lips-12*, *lips-13*) that may participate in fat metabolism. As we previously found in hermaphrodites (*Shi and Murphy, 2014*), mating decreases fat storage in males (*Figure 5B*).

Surprisingly, vitellogenins (*vit-4*, *vit-3*, *vit-5*, *vit-6*, *vit-2*), which encode yolk protein precursors made in the female/hermaphrodite intestine for transport into developing oocytes (*Kimble and Sharrock, 1983*) and as such are not normally expressed in males (*Figure 4G*, left), were the top up-regulated genes in mated males, expressed on average 19 times higher in mated males than in solitary unmated males (*Supplementary file 3*). Overproduction of vitellogenins has been shown to be deleterious for hermaphrodites: vitellogenins accumulate in the head and body of older hermaphrodites (*Garigan et al., 2002*); long-lived insulin signaling mutants repress *vit* gene expression (*Murphy et al., 2003*; *Seah et al., 2016*); overexpression of vitellogenin reduces the lifespan of long-lived mutants (*Seah et al., 2016*); and knockdown of the *vit* genes in wild-type hermaphrodites extends lifespan (*Murphy et al., 2003*; *Seah et al., 2016*). Mating induced ectopic expression of

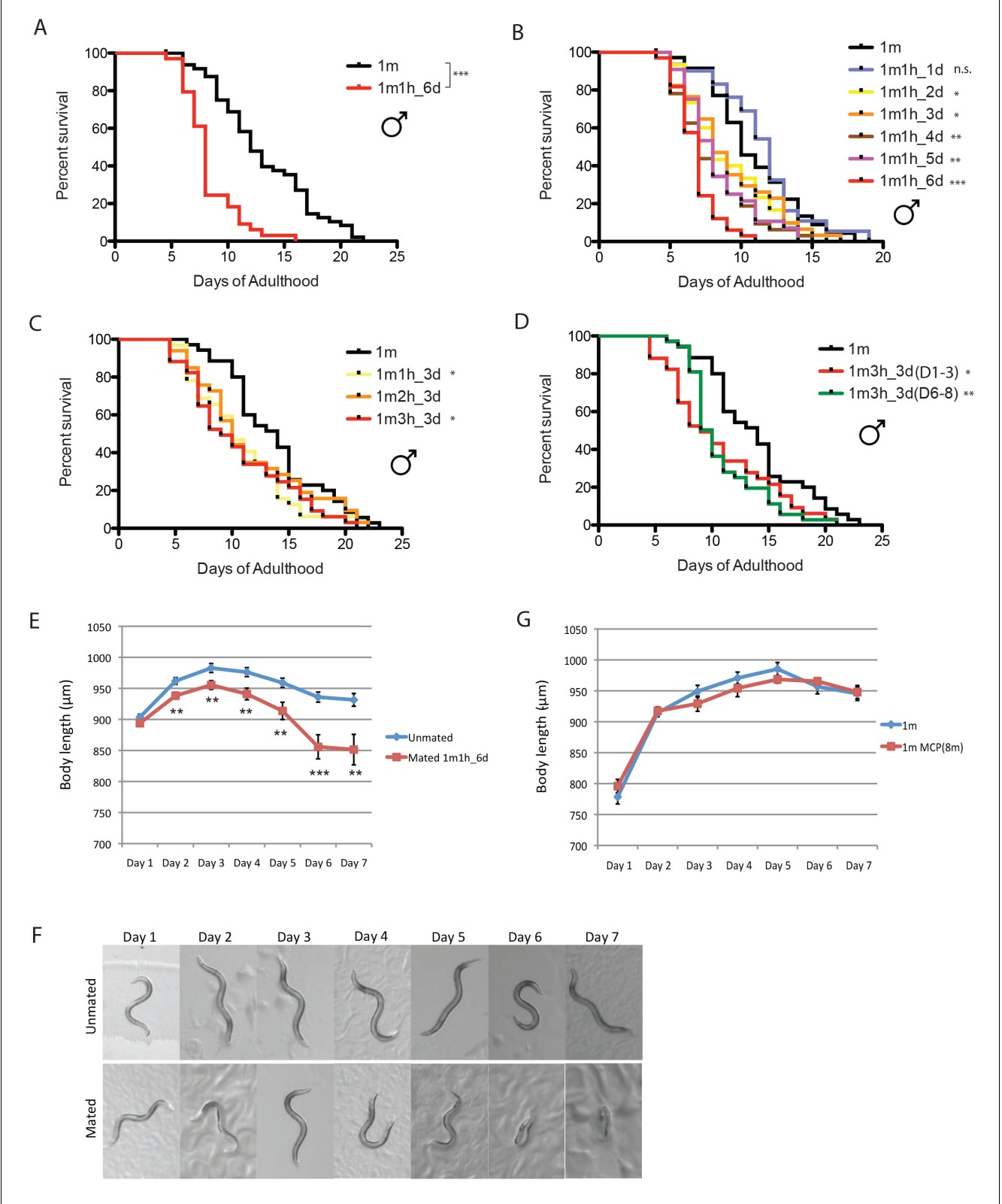

**Figure 3.** *C. elegans* males shrink and die early after mating. (**A**) Lifespans of unmated solitary and mated *fog-2(q71)* males. Solitary males: 13.1 ± 0.6 days, n = 50; mated males: 8.3 ± 0.4 days, n = 34, p<0.0001. Each male was paired with a *fog-2(q71)* hermaphrodite on a single 35 mm plate during Day 1–6 of adulthood. Unless noted, all the hermaphrodites used are *fog-2(q71)*. (**B**) Male post-mating lifespan decrease is mating duration-dependent: Unmated solitary males: 10.9 ± 0.6 days, n = 35; one male and one hermaphrodite mating on Day 1 of adulthood: 11.4 ± 0.6 days, n = 31, p=0.3697;

*Figure 3 continued on next page*

*Figure 3 continued*

mating from Day 1–2: 9.0 ± 0.6 days, n = 30, p=0.0325; mating from Day 1–3: 9.1 ± 0.6 days, n = 34, p=0.0452; mating from Day 1–4: 7.9 ± 0.5 days, n = 32, p=0.0002; mating from Day 1–5: 8.3 ± 0.4 days, n = 34, p=0.0006; mating from Day 1–6: 6.8 ± 0.3 days, n = 33, p<0.0001. (C) Lifespans of one male paired with different number of hermaphrodites during Day 1–3 of adulthood: solitary unmated males: 13.8 ± 0.7 days, n = 35; one male with one hermaphrodite: 10.8 ± 0.6 days, n = 32, p=0.0175; one male with two hermaphrodites: 11.6 ± 0.9 days, n = 33, p=0.1435; one male with three hermaphrodites: 10.6 ± 0.8 days, n = 34, p=0.0147. (D) Lifespans of one male paired with three hermaphrodites for 3 days but at different time of adulthood. Solitary unmated males: 13.8 ± 0.7 days, n = 35; mating during Day 1–3 of adulthood: 10.6 ± 0.8 days, n = 34, p=0.0147; mating during Day 6–8 of adulthood: 10.8 ± 0.6 days, n = 37, p=0.0022. (E) Length of unmated and mated *fog-2* males: t-test, **p<0.01, ***p<0.001. (F) Representative pictures of the same unmated solitary male and male paired with one hermaphrodite from Day 1-Day 6 of adulthood. (G) Male pheromone does not induce body shrinking. Length of solitary *fog-2* males on plates conditioned by eight wild-type males.

VIT-2::GFP in the anterior intestine of males, confirming our gene expression data. Such expression induction was germline function-dependent, as FUdR treatment of males blocked VIT-2::GFP expression in mated males (*Figure 4G*, *Figure 4—figure supplement 2D*).

In addition to the increase in *vit* gene expression, the binding motif for UNC-62, a master transcriptional regulator of *vit* genes in hermaphrodites (*Van Nostrand et al., 2013*), emerged from unbiased motif analysis of the up-regulated gene set (*Figure 4—figure supplement 2C*). Using RNAi, we found that knocking down *unc-62* was sufficient to rescue the lifespan decrease in mated males (*Figure 4H*). Thus, the mis-expression of vitellogenins upon mating contributes to post-mating death in males. The DAE (DAF-16 Associated Element) motif is present in most *vit* gene promoters, which are also Class 2 DAF-16 genes (*Murphy et al., 2003*). The DAE is bound by PQM-1, a transcription factor that is involved in the regulation of multiple processes, including development, stress response, metabolism, and longevity (*Tepper et al., 2013*; *Dowen et al., 2016*). We found that mated *pqm-1(ok485)* deletion males lived as long as unmated controls (*Figure 4I*), suggesting that PQM-1 is also required for post-mating male death.

## Mating-induced death and pheromone-induced killing use distinct molecular mechanisms

To determine whether pheromone-dependent killing, which like mating-induced death requires the germline, utilizes the same downstream mechanisms to kill males, we performed genome-wide transcriptional analysis of worms under conditions where they are exposed to high levels of male pheromone and have short lifespans (*Figures 1G* and *2C*): (1) grouped *daf-22* males on plates conditioned by wild-type males vs. grouped *daf-22* males on control plates, and (2) grouped vs. solitary neuronally-masculinized hermaphrodites (*Supplementary file 4*). Three main Gene Ontology groups emerged from these comparisons: innate immunity and defense responses, metal ion/cadmium response, and glycoprotein metabolism. Notably, this pattern of gene expression was very different from that of mated males (*Figure 5A*, *Figure 5—figure supplement 1*; Pearson correlation = −0.0594 for the whole transcriptome). Specifically, upregulation of vitellogenin genes was not a signature of male pheromone-treated animals (*Figure 5—figure supplement 1C*), also supporting the notion that the two pathways act through distinct molecular mechanisms. While fat is reduced in males after mating (*Figure 5B*), MCP treatment caused no significant changes in fat metabolism gene expression (*Figure 5—figure supplement 1A*) or Oil Red O staining (*Figure 5C*). We showed previously that osmotic stress resistance correlates well with shrinking in mated hermaphrodites, whereas fat loss does not account for such shrinking (*Shi and Murphy, 2014*). Likewise, we found that mated wild-type worms lost about 30% of their glycogen stores post-mating in a germline-dependent manner (*Figure 5D*). The mating-induced glycogen storage decrease and subsequent shrinking is conserved between sexes (*Figure 5—figure supplement 2*), while glycogen stores are not affected by male pheromone (*Figure 5E*).

Previously identified gene expression patterns of longevity pathways did not appear in our analysis (*Tepper et al., 2013*; *Lakhina et al., 2015*), suggesting a novel pathway for lifespan shortening in the presence of male pheromone. However, comparison of the list of genes significantly up-regulated upon MCP treatment of *daf-22* males (*Supplementary file 5*) to previously published arrays of male pheromone-treated hermaphrodites (*Maures et al., 2014*) yielded a significant overlap (p-value=3.05E-06), including *ins-11*; *ins-11* mutants are partially resistant to death by male pheromone

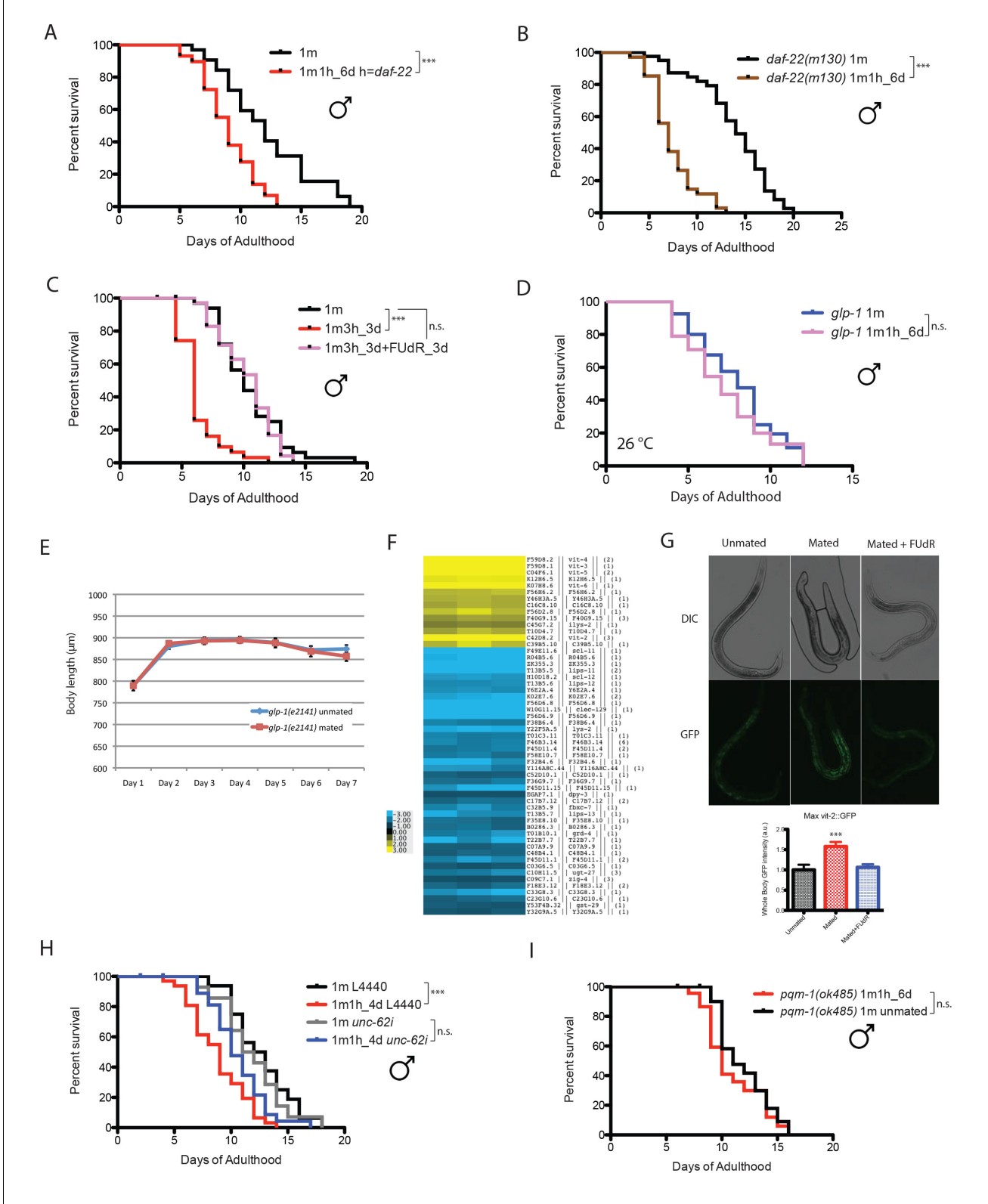

**Figure 4.** Mating-induced death is germline-dependent. (**A**) Lifespans of *fog-2* males mated with *daf-22(m130)* hermaphrodites. Unmated solitary *fog-2* males: 12.1 ± 0.6 days, n = 32; mated males: 9.0 ± 0.4 days, n = 29, p=0.0001. In the mated group, one *fog-2(q71)* male was paired with one *daf-22 (m130)* hermaphrodite from Day 1- Day 6 of adulthood. (**B**) Lifespans of unmated and mated *daf-22(m130)* males. Unmated solitary *daf-22(m130)* males: 13.8 ± 0.6 days, n = 40; mated *daf-22(m130)* males: 7.4 ± 0.4 days, n = 34, p<0.0001. In the mated group, one *daf-22(m130)* male was paired with one

*Figure 4 continued on next page*

*Figure 4 continued*

*fog-2(q71)* hermaphrodite from Day 1- Day 6 of adulthood. (**C**) FUdR can rescue male post-mating early death. Unmated solitary males: 10.5 ± 0.5 days, n = 35; one male with three hermaphrodites for three days: 6.4 ± 0.3 days, n = 31, p<0.0001; one male with three hermaphrodites for three days but in the presence of 50 μM FUdR during the three days' mating: 10.2 ± 0.4 days, n = 36, p=0.7086 (compared with unmated solitary group). (**D**) Lifespans of unmated and mated *glp-1(e2141)* males: unmated solitary *glp-1* males: 8.0 ± 0.4 days, n = 40; mated *glp-1* males: 7.2 ± 0.4 days, n = 40, p=0.3178. The assay was performed at 26°C, in mated group, one *glp-1* male was paired with one *fog-2* hermaphrodite from Day 1–6. (**E**) Length of mated and unmated *glp-1(e2141)* males. (The same population as in ***Figure 4D***.). (**F**) Expression heatmap of genes whose expression is significantly changed in mated males based on SAM analysis. (**G**) Ectopic expression of VIT-2::GFP in mated males is germline-dependent. 5 days' mating, pictures were taken on Day 6 of adulthood. Representative images are shown above the quantification of VIT-2::GFP expression [maximum ± SE (error bars)], a.u., arbitrary units. ***p<0.001, t-test. (**H**) *unc-62* RNAi suppresses male post-mating early death. Unmated solitary male on L4440: 12.6 ± 0.7 days, n = 25; mated males on L4440: 8.8 ± 0.5 days, n = 33, p=0.0001. Unmated males on *unc-62* RNAi: 11.9 ± 0.8 days, n = 25; mated males on *unc-62* RNAi: 10.6 ± 0.5 days, n = 34, p=0.1249 (compared to unmated males on *unc-62* RNAi). (**I**) *pqm-1(ok485)* mated males have similar lifespans as unmated controls. Unmated solitary *pqm-1(ok485)* males: 11.9 ± 0.5 days, n = 25; mated *pqm-1(ok485)* males: 11.0 ± 0.6 days, n = 29, p=0.2782. In the mated group, one *pqm-1(ok485)* male was paired with one *fog-2(q71)* hermaphrodite from Day 1- Day 6 of adulthood.

The following figure supplements are available for figure 4:

**Figure supplement 1.** Mating-induced lifespan decrease is germline-dependent.

**Figure supplement 2.** Microarray analysis of mated males.

---

(***Maures et al., 2014***). These results suggest that some mechanisms of male-pheromone-induced death are shared between the sexes, but may act independently of known longevity pathways.

Unbiased motif analysis revealed that DAE (DAF-16 Associated Element) motif was also enriched in these male pheromone-treated conditions (***Figure 5—figure supplement 3***). We found that *pqm-1(ok485)* null mutant males were not short-lived when treated with male pheromone (eight males per plate for conditioning; ***Figure 5F***), suggesting that male pheromone killing, like mating-induced death, is mediated at least in part by PQM-1.

Both male pheromone-induced killing and mating-induced death require the germline; to examine possible mechanisms of germline effects, we performed DAPI staining of males in pheromone-treated and mated conditions. In a fraction of Day four unmated solitary males, the transition zone (marked by crescent shaped nuclei) disappeared (***Figure 5G***, ***Figure 5—figure supplement 4***), and the meiotic region expanded to the distal arm, evidenced by the presence of sperm before U-shaped turn (see ***Figure 5—figure supplement 4***), indicating the loss of mitotic proliferating cells in the germline. Shrinking of the transition zone was observed significantly less in mated animals, but no significant differences were apparent in MCP-treated males (***Figure 5G***). Therefore, our results suggest that mating causes an increase in the number of mitotically proliferating cells, whereas an unknown signal from the germline, rather than an obvious morphological change, may be responsible for the lifespan shortening effects of male pheromone.

## Mating-induced male death is evolutionarily conserved and unavoidable

Previously, we found that *C. remanei* females, like *C. elegans* hermaphrodites, shrink and die faster after mating (***Shi and Murphy, 2014***), suggesting that these mechanisms are evolutionarily conserved in females. We found that *C. remanei* males also lived significantly shorter after mating with a female *C. remanei* for 6 days (***Figure 6A***). While female death requires successful cross-progeny production (***Shi and Murphy, 2014***), *C. elegans* males died early when mated with a *C. remanei* female for 6 days (***Figure 6B***), even though no cross-progeny result from this mating. This result, together with the germline dependence of mating-induced death, suggests the process of mating and upregulation of male germline activity is sufficient to induce death, regardless of the species of the recipient female.

## Gonochoristic species are immune to male pheromone toxicity

Because *C. elegans* is an androdioecious (males and hermaphrodites) species, we wondered whether male pheromone-mediated killing also occurs in a true 50/50 male/female (gonochoristic) species, where one would expect the level of male pheromone to be high under normal conditions. When

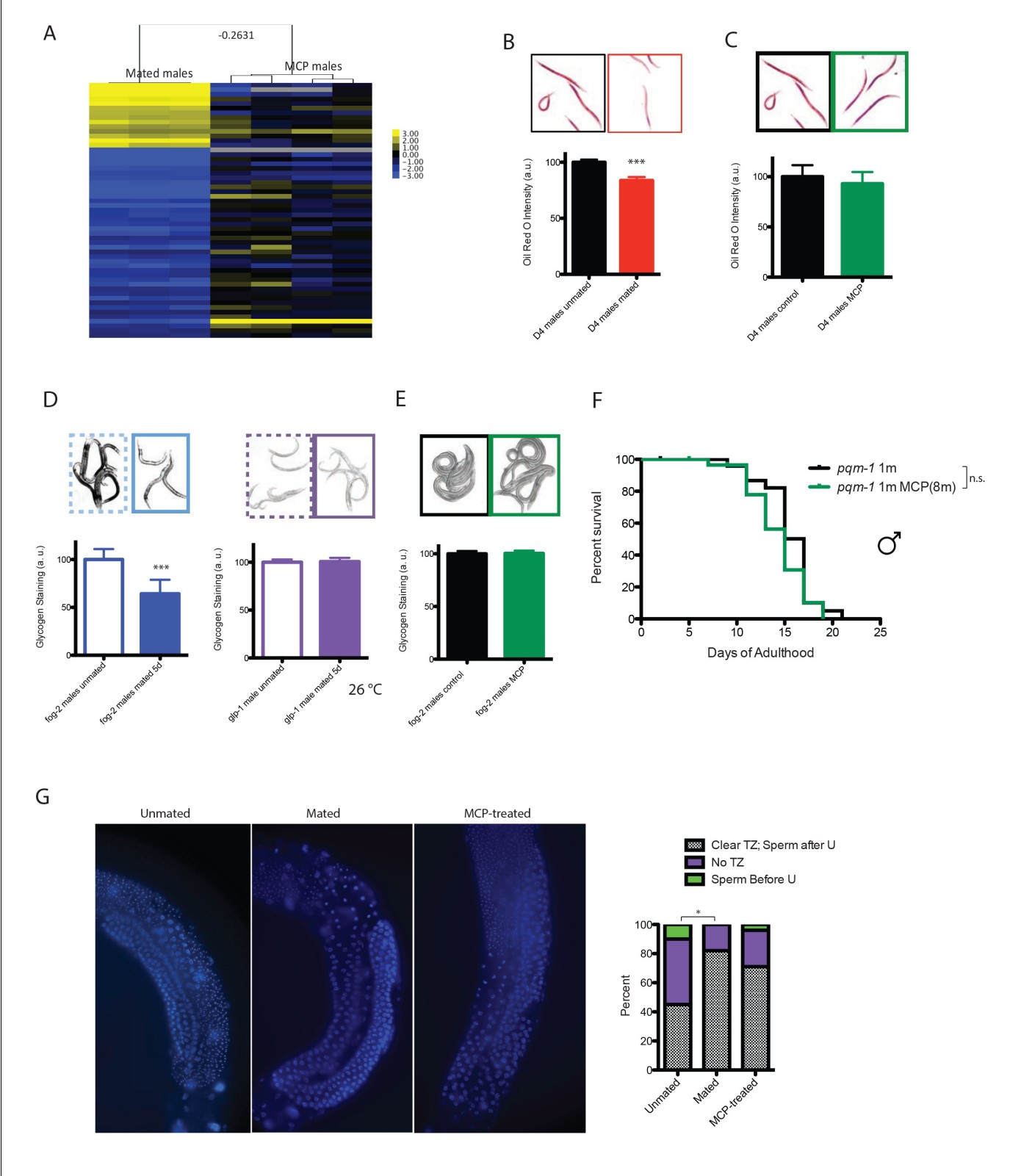

**Figure 5.** Mating-induced and male pheromone-induced death are distinct. (**A**) Transcriptional profiles of mated males and MCP-treated males are different. Heatmap cluster of mated males (left) and MCP-treated vs untreated grouped *daf-22* males (right); Pearson correlation = −0.27. The cluster only contains genes with significant changes in mated males by SAM, 0% FDR. (**B**) Mating induces significant fat loss in males. Representative pictures of Oil red O staining are shown above the quantitation. Males lost about 20% of their fat after mating on Day 4, p<0.001. Error bars represent SE. (**C**)

*Figure 5 continued on next page*

*Figure 5 continued*

Male pheromone exposure fails to induce fat loss in males. Four days' MCP treatment. Representative pictures of Oil red O staining are shown above the quantitation. Unconditioned control males are framed by black lines, and MCP-treated males are framed by green lines. (**D**) Glycogen staining of mated and unmated males. Left: mated *fog-2* (wt) males lost over 30% glycogen after 5 days' mating; \*\*\*p<0.001. Right: mated *glp-1* males did not lose glycogen after mating. The staining intensity was normalized to unmated males of each genotype. Representative pictures are shown above the quantitation. Unmated males are framed by dashed lines, and mated males are framed by solid lines. (**E**) No glycogen loss after male pheromone exposure. Four days' MCP treatment. Representative pictures of iodine staining are shown above the quantitation. Unconditioned control males are framed by black lines, and MCP-treated males are framed by green lines. (In **D–E**, error bars represent SD.). (**F**) Loss of the DAE-dependent transcription factor PQM-1 suppresses male pheromone-induced death. Lifespans of control solitary *pqm-1(ok485)* males: 15.6 ± 0.6 days, n = 25; solitary *pqm-1(ok485)* males on plates conditioned by eight males: 14.4 ± 0.6 days, n = 25, p=0.1627. (**G**) DAPI staining of Day six males' germlines (left). Right: categorical quantification of germline morphology: mating causes more obvious change in male germline morphology than male pheromone does. TZ: transition zone; U: U-shaped turn of male germline. See *Figure 5—figure supplement 4* for details.

The following figure supplements are available for figure 5:

**Figure supplement 1.** Transcriptional profiles of mated and male pheromone-induced males are distinct.

**Figure supplement 2.** Glycogen staining of mated vs unmated hermaphrodites and males.

**Figure supplement 3.** Enriched motifs of male pheromone-induced transcriptional changes.

**Figure supplement 4.** Germline of mated and MCP-treated males.

---

exposed to low levels of *C. remanei* male pheromone (eight males per plate for conditioning), neither *C. remanei* males nor females were short-lived (*Figure 6C,D*). At a higher concentration (30 males per plate for conditioning), multiple trials of *C. remanei* females and males on male-conditioned plates failed to reveal any sensitivity to either *C. remanei* or *C. elegans* male pheromone (*Figure 6E,F*), in contrast to previous reports in which males and females were grouped (*Maures et al., 2014*); this result suggests that the lifespan shortening in the latter study was caused by mating rather than by male pheromone.

Interestingly, we found that male pheromone toxicity can act across species: *C. elegans* hermaphrodites died early when exposed to *C. remanei* male pheromone (*Figure 6G*), suggesting that the difference in sensitivity to male pheromone might stem from the *perception* of pheromone as a toxin, rather than from toxicity of the pheromone itself. *C. remanei* can sense pheromone, but uses it to distinguish potential partners and competitors (*Figure 6H*), rather than to kill males. The differential sensitivity to male-pheromone-induced killing between *C. elegans* males and hermaphrodites also suggests that the latter might only be experienced as an off-target effect under extremely high male pheromone conditions (*Figure 6I*).

To determine whether the differences in male-pheromone-induced killing are a more general phenomenon, we examined the effect of male pheromone on other androdioecious and gonochoristic species. Like *C. remanei* males, two other gonochoristic species, *C. brenneri* and *C. nigoni*, are also immune to male pheromone killing (*Figure 7A,B*). By contrast, males from two evolutionarily distant androdioecious species, *C. briggsae* and *C. tropicalis*, died significantly earlier when exposed to male pheromone (*Figure 7C,D*), just as *C. elegans* males do. These results strongly indicate that male pheromone-dependent killing is shared by males of androdioecious species.

## Male pheromone reduces male offspring

The fact that male pheromone selectively kills males of multiple, independently-evolved androdioecious species suggests a role for male pheromone killing in these populations, but the lifespan effects we observed to this point are largely post-reproductive, arguing against any reproductive selection under these conditions. However, those experiments were designed specifically to probe adult phenotypes rather than development or mating, by only applying pheromone to adult worms. In order to better mimic conditions in which worms would be exposed to male pheromone their entire lives, we placed *fog-2* eggs on male pheromone conditioned plates, and measured developmental rates, lifespan, mating rates, and brood size. Other than a slight difference at 42 hr that

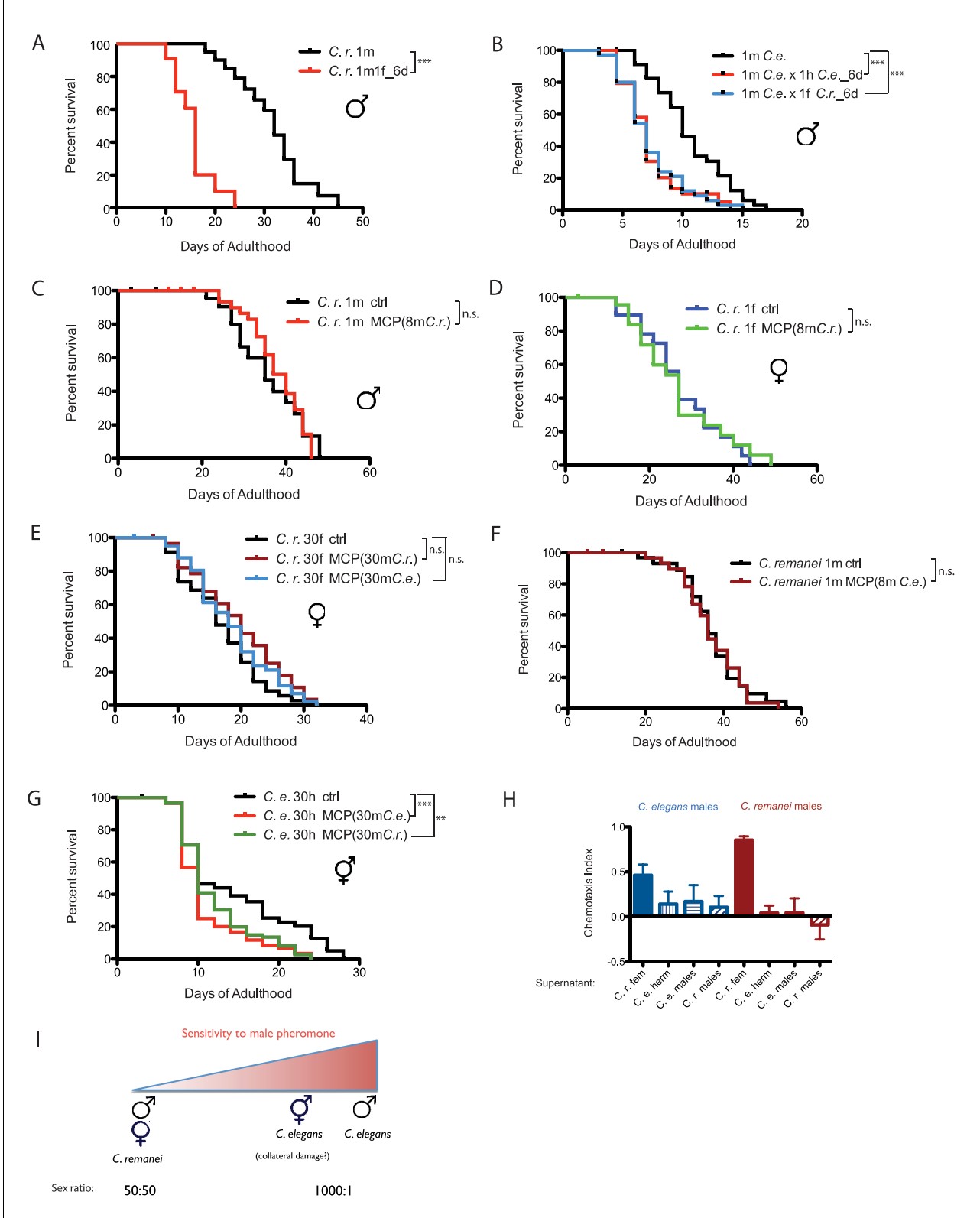

**Figure 6.** Mating-induced death is evolutionarily conserved, whereas male pheromone-induced death is not. (**A**) Mated *C. remanei* males also live shorter. Unmated solitary *C. remanei* males: 31.4 ± 1.7 days, n = 72; mated *C. remanei* males: 15.7 ± 1.2 days, n = 28, p<0.0001. In mated group: one *C. remanei* male was paired with one *C. remanei* female from Day 1-Day 6 of adulthood. (**B**) Lifespans of *C. elegans* males mated with *C. elegans* hermaphrodites and *C. remanei* females. Unmated solitary *C. elegans* males: 10.2 ± 0.6 days, n = 35; *C. elegans* males mated with *C. elegans*

*Figure 6 continued on next page*

*Figure 6 continued*

hermaphrodites: 7.4 ± 0.4 days, n = 35, p=0.0001; *C. elegans* males mated with *C. remanei* females: 7.4 ± 0.4 days, n = 35, p=0.0003. In mated groups, one *C. elegans* male was paired with either one *C. elegans* hermaphrodite or one *C. remanei* female from Day 1–6 of adulthood. (C) Lifespans of solitary *C. remanei* males on plates conditioned by eight *C. remanei* males. Solitary *C. remanei* males on control plates: 35.8 ± 2.0 days, n = 34; solitary *C. remanei* males on male-conditioned plates: 37.8 ± 1.2 days, n = 34, p=0.8501. (D) Lifespans of solitary *C. remanei* females on plates conditioned by eight *C. remanei* males. Solitary *C. remanei* females on control plates: 27.6 ± 2.2 days, n = 24; solitary *C. remanei* females on male-conditioned plates: 27.0 ± 2.5 days, n = 30, p=0.8306. (E) Lifespans of grouped *C. remanei* females on plates conditioned by 30 males. *C. remanei* females on control plates: 15.8 ± 0.9 days, n = 60; *C. remanei* females on plates conditioned by *C. remanei* males: 19.5 ± 1.3 days, n = 30, p=0.0636; *C. remanei* females on plates conditioned by *C. elegans fog-2* males: 18.5 ± 0.9 days, n = 60, p=0.1770. (F) Lifespans of solitary *C. remanei* males on plates conditioned by eight *C. elegans* males. Solitary *C. remanei* males on control plates: 37.2 ± 1.7 days, n = 40; solitary *C. remanei* males on *C. elegans* male-conditioned plates: 36.7 ± 1.4 days, n = 38, p=0.7774. (G) Lifespans of grouped *C. elegans fog-2* hermaphrodites on plates conditioned with 30 males. *fog-2* hermaphrodites control: 14.4 ± 0.8 days, n = 90. *fog-2* hermaphrodites on plates conditioned by *fog-2* males: 10.9 ± 0.6 days, n = 60, p=0.0004; *fog-2* hermaphrodites on plates conditioned by *C. remanei* males: 11.9 ± 0.5 days, n = 90, p=0.0042. (H) Chemotaxis of *C. elegans* (left, blue) and *C. remanei* (right, red) to supernatants from *C. elegans* males, *C. remanei* males, *C. elegans* N2 hermaphrodites, and *C. remanei* females. See Materials and methods for detailed description. *C. e.* males to supernatant of *C. r.* females: Chemotaxis Index (CI) is 0.46 ± 0.11 (mean ± SEM, n = 12 [plates]); *C. e.* males to supernatant of *C. e.* hermaphrodites: CI = 0.14 ± 0.13 (n = 10); *C. e.* males to supernatant of *C. e.* males: CI = 0.17 ± 0.17 (n = 12); *C. e.* males to supernatant of *C. r.* males: CI = 0.11 ± 0.12 (n = 11); *C. r.* males to supernatant of *C. r.* females: CI = 0.85 ± 0.04 (n = 12); *C. r.* males to supernatant of *C. e.* hermaphrodites: CI = 0.04 ± 0.08 (n = 12); *C. r.* males to supernatant of *C. e.* males: CI = 0.04 ± 0.15 (n = 12); *C. r.* males to supernatant of *C. r.* males: CI = −0.09 ± 0.16 (n = 12). (I) Sensitivity to male pheromone-induced lifespan reduction. *C. elegans* males are the most sensitive to male pheromone-induced killing, whereas both *C. remanei* sexes are immune to this effect.

disappeared by 48 hr, we observed no significant effects of MCP on the developmental rates of either males or hermaphrodites (*Figure 8—figure supplement 1*). By contrast, male pheromone conditioning from egg onward caused a severe (36%) shortening of lifespan (*Figure 8A*, *Figure 8— figure supplement 2A*). Moreover, male pheromone significantly decreased male fertility (*Figure 8B*, *Figure 8—figure supplement 2B*). (Note that male pheromone-induced male fertility decrease is distinct from defects in male mating that arise with age (*Chatterjee et al., 2013*); we observed similar male fertility decline with age in control animals, but the males treated with male pheromone from egg onward exhibited an additional fertility decline compared with age-matched control males.) Finally, male pheromone treatment decreases the number of progeny produced by those animals who do successfully mate (*Figure 8C*). By contrast, male pheromone treatment did not affect the brood size of self-fertilized hermaphrodites (*Figure 8D*). These results suggest that exposure to male pheromone during early life specifically reduces male fertility.

## Discussion

### *Caenorhabditis* males die early from two independent phenomena: male pheromone-mediated killing and mating-induced death

Here we found that male pheromone killing is the major cause of population density-dependent life-span decrease in *C. elegans* males, and is only utilized by androdioecious *Caenorhabditis* males. By contrast, all sexes of *Caenorhabditis* species succumb to mating-induced death, while both sexes of gonochoristic *Caenorhabditis* species are immune to male pheromone toxicity. The toxicity of male pheromone may explain the contradictory results from previous publications in which grouped males were used as the control in testing whether mating affects the lifespan of *Caenorhabditis* males (*Gems and Riddle, 1996*; *Van Voorhies, 1992*). Masculinization of neurons in hermaphrodites not only increases their sensitivity to male pheromone, but also is sufficient to induce the production of male-like toxic pheromone, suggesting that neurons play two major and distinct roles in this type of killing. The germline and the DAE-dependent transcription factor PQM-1 are required for both mat-ing-induced and male pheromone-mediated death, but the downstream expression changes upon mating and pheromone treatment are distinct; only mating induces *vitellogenin* gene expression in males and causes shrinking. Thus, we have discovered two distinct mechanisms that accelerate aging in *Caenorhabditis* males.

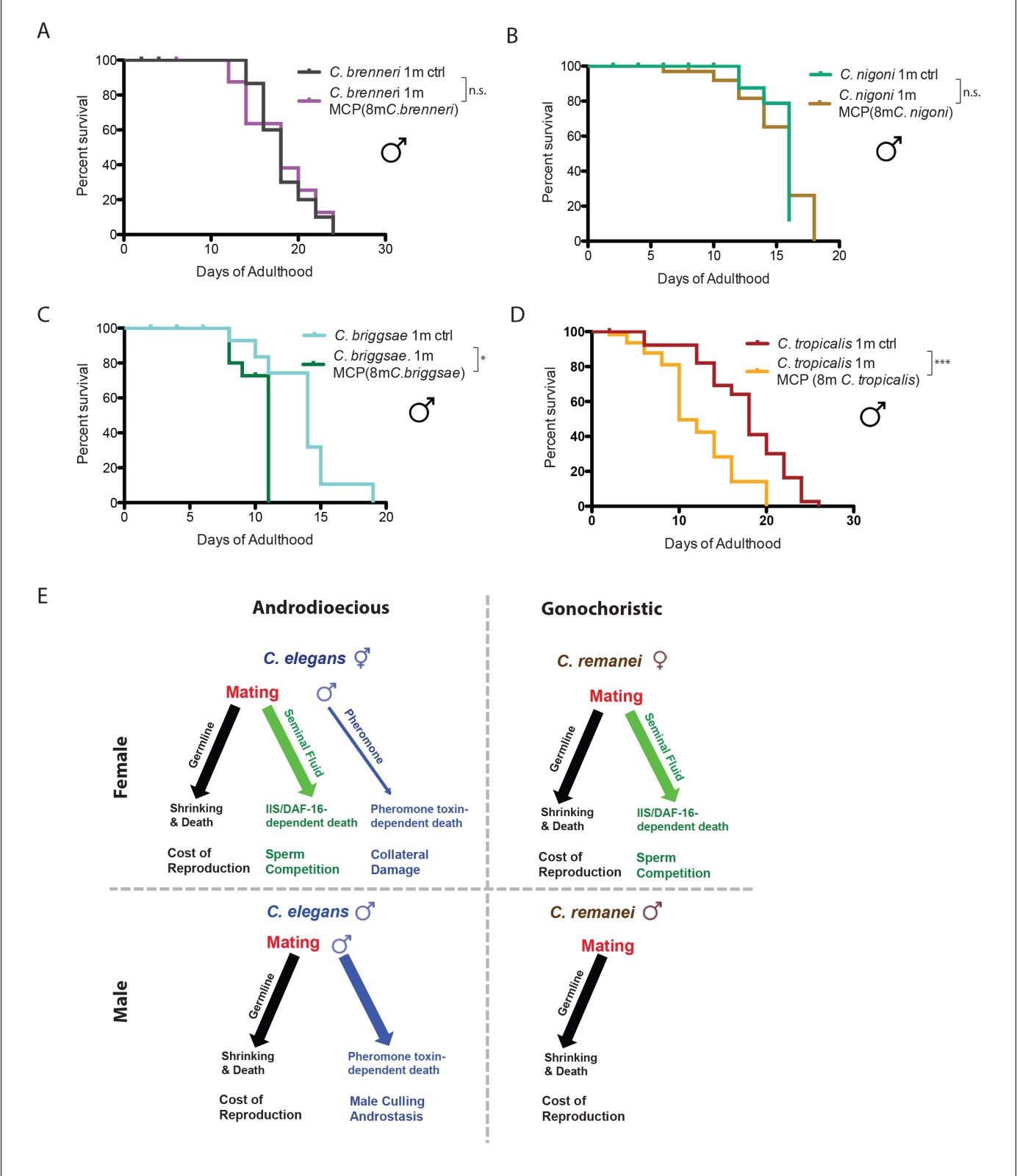

**Figure 7.** Gonochoristic species are immune to male pheromone killing; androdioecious species are susceptible. (**A**) Lifespans of solitary *C. brenneri* males on plates conditioned by 8 *C. brenneri* males. Solitary *C. brenneri* males control: 18.1 ± 0.8 days, n = 33; solitary *C. brenneri* males on MCP: 17.8 ± 1.1 days, n = 32, p=0.9915. (**B**) Lifespans of solitary *C. nigoni* males on plates conditioned by 8 *C. nigoni* males. Solitary *C. nigoni* males control: 15.3 ± 0.4 days, n = 32; solitary *C. nigoni* males on MCP: 15.2 ± 0.6 days, n = 40, p=0.7443. (**C**) Lifespans of solitary *C. briggsae* males on plates

*Figure 7 continued on next page*

*Figure 7 continued*

conditioned by 8 *C. briggsae* males. Solitary *C. briggsae* males control: 13.7 ± 0.8 days, n = 38; solitary *C. briggsae* males on MCP: 10.3 ± 0.3 days, n = 54, p=0.0192. (D) Lifespans of solitary *C. tropicalis* males on plates conditioned by 8 *C. tropicalis* males. Solitary *C. tropicalis* males control: 17.7 ± 0.8 days, n = 40; solitary *C. tropicalis* males on MCP: 12.2 ± 1.0 days, n = 60, p=0.0002. (E) Model of the effects of mating and male pheromone on androdioecious and gonochoristic female and males. *C. elegans* hermaphrodites (upper left); *C. remanei* females (upper right); *C. elegans* males (lower left); *C. remanei* males (lower right).

## Germline activation induces deleterious changes that kill all sexes in *Caenorhabditis* species

*C. elegans* males and hermaphrodites share many post-mating changes. As we found previously for mated females and hermaphrodites (*Shi and Murphy, 2014*), *Caenorhabditis* males also experience germline-dependent shrinking, glycogen loss, and death after mating. Germline up-regulation also leads to ectopic expression of vitellogenins, which contributes to the post-mating lifespan decrease in males. Previously, these yolk protein precursors were only reported to be expressed in hermaphrodites, where vitellogenin proteins are taken up by oocytes; vitellogenin production in males does not have an obvious purpose. Mating also induces significant overexpression of *vit* genes in hermaphrodites (*DePina et al., 2011*), indicating that vitellogenin expression is closely coupled with mating-induced germline up-regulation in both sexes. Such coupling may be strong enough to overcome the normal repression of male vitellogenin expression. Germline-dependent body shrinking, glycogen loss, and ectopic vitellogenin expression contribute to male post-mating death, which is conserved between the sexes. The striking similarity of germline-dependent post-mating changes in *Caenorhabditis* males and females suggests that this mechanism is largely conserved between sexes, and may represent an unavoidable cost of reproduction as a result of mating.

Germline-dependent lifespan shortening appears to be conserved across species over large evolutionary distances, as it occurs in all *Caenorhabditis* species we tested. Male post-mating death is also conserved beyond the *Caenorhabditis* genus, as *Drosophila* males die earlier after mating, as well (*Partridge and Farquhar, 1981*). It was previously noted that the lifespan of Korean eunuchs was significantly longer than the lifespan of non-castrated men with similar socio-economic status (*Min et al., 2012*), analogous to the long lifespan of germline-less *C. elegans* (*Hsin and Kenyon, 1999*) and *Drosophila* (*Flatt et al., 2008*), while the significantly (35%) shortened lifespan of Chinese emperors who were noted to be particularly promiscuous might be an example of the opposite effect on the germline (*Shi et al., 2015*), suggesting that some aspects of germline-dependent male post-mating death may be conserved across great evolutionary distances.

## Male pheromone-induced killing may be a strategy to selectively reduce the male population

*C. elegans* are subject to killing by male pheromone, while *C. remanei* are not. Our cross-species results suggest that *C. remanei* male pheromone is perceived as a toxin by *C. elegans*, but *C. remanei* are immune to both *elegans* and *remanei* pheromone (*Figure 6C–F*). The preponderance of males in a 50:50 population, as in the case of *C. remanei*, makes the use of pheromone as a toxin less likely, as it would cause too much off-target death to be useful for sperm competition purposes. The toxic effect of pheromone may not be due to the pheromone itself, but rather to a perception of pheromone as a toxin, with a greater effect in males than in hermaphrodites. Hermaphrodite death at high male pheromone concentration (*Maures et al., 2014*)—which might happen rarely in nature – might simply be collateral damage, as hermaphrodites are far less sensitive than males to male pheromone (*Figure 6I*) toxin. The lack of significant changes observed in the developmental rates of either males or hermaphrodites, or on the brood size of hermaphrodites, indicates that the primary effect of male pheromone might be on male reproductive capacity.

*Caenorhabditis* species might utilize pheromones in such different ways due to their different modes of reproduction. Androdioecious species males do not appear to use pheromones efficiently as chemical messengers to facilitate mating, since they are less able to distinguish hermaphrodites' pheromone from other species' female or male pheromone; in fact, *C. elegans* males are slightly attracted to their own male pheromone, in part explaining their clumping (*Chasnov et al., 2007*) (*Figure 6H*), despite the fact that male pheromone is very toxic to *C. elegans* males. In the

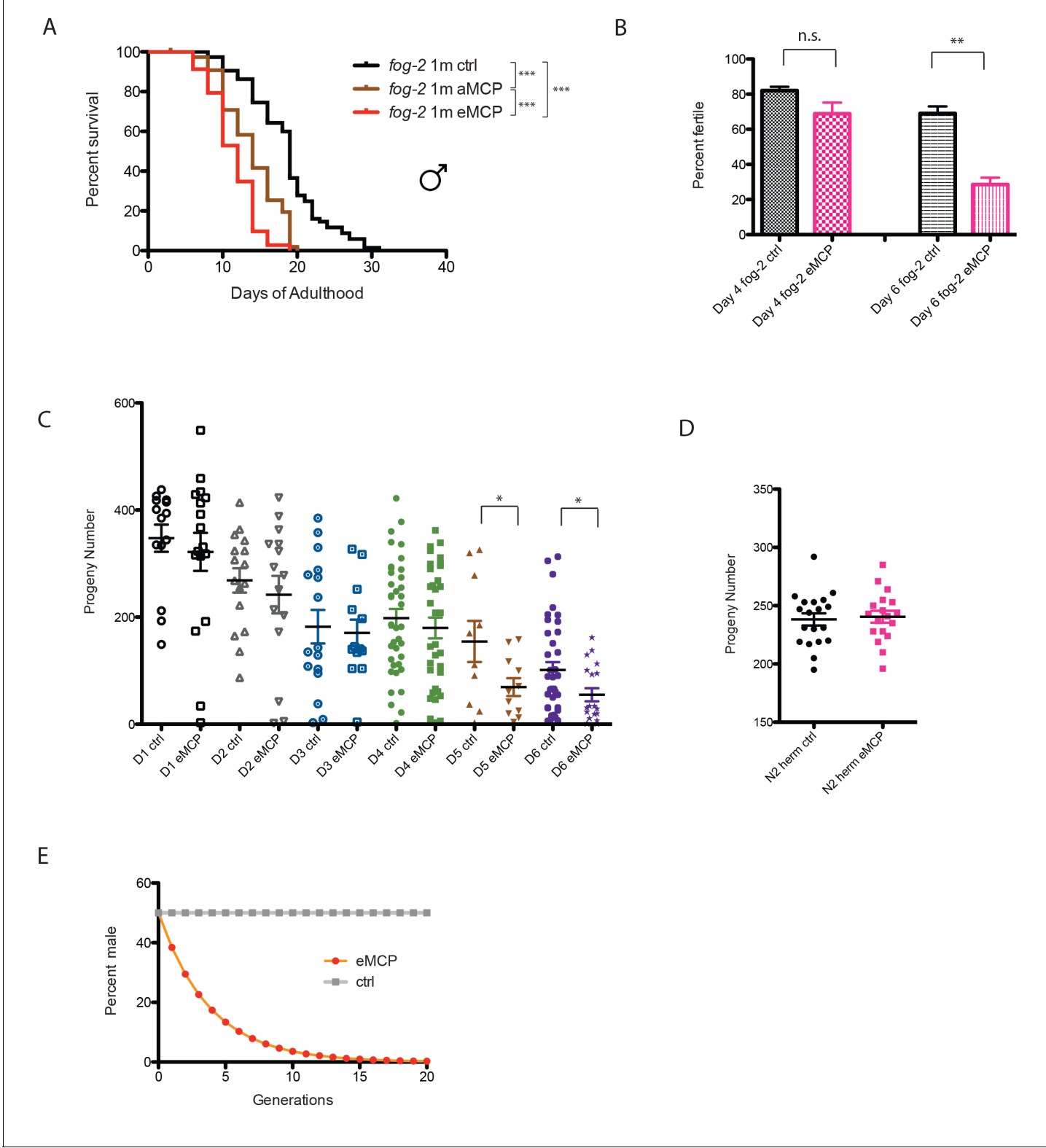

**Figure 8.** Male pheromone reduces male offspring. (**A**) Male pheromone conditioning from egg onward causes a more severe lifespan shortening. (Pooled results from two independent assays. See *Figure 8—figure supplement 2* for results of separate lifespan assays, which were also significant.) Solitary *C. elegans fog-2* males control: 18.5 ± 0.6 days, n = 75; solitary *C. elegans fog-2* males on MCP from early adulthood onward (aMCP): 13.9 ± 0.4 days, n = 79, p<0.0001 (compared to the control); solitary *C. elegans fog-2* males on MCP from egg onward (eMCP): 11.4 ± 0.4 days, n = 80, p<0.0001
*Figure 8 continued on next page*

*Figure 8 continued*

(compared to the control), p<0.0001 (compared to aMCP). (**B**) Male pheromone decreases *fog-2* male fertility on Day 6. Male treatment started from egg onward. Each male was paired with one virgin *fog-2* hermaphrodite at indicated time for 24 hr. On Day 4 of adulthood, the percent of males who were able to fertilize Day 1 virgin *fog-2* hermaphrodites: three biological replicates: ctrl: 82 ± 2%; MCP: 69 ± 6%, p=0.12, unpaired t-test. By Day 6 of adulthood, the percent of males who were able to fertilize Day one virgin *fog-2* hermaphrodites (three biological replicates) had significantly decreased: ctrl: 69 ± 4%; MCP: 29 ± 4%, p=0.0019. (**C**) Male pheromone treatment decreases the number of progeny produced by those animals who do successfully mate. The difference appears by Day 5. See *Figure 8—figure supplement 3* for detailed numbers. *p<0.05, unpaired t-test. (**D**) Brood size of self-fertilized N2 hermaphrodites is not affected by male pheromone. Ctrl: 238.2 ± 5, n = 19; MCP-treated: 240.5 ± 5, n = 18, p=0.7571, unpaired t-test. (**E**) Theoretical calculation of male pheromone's effect on male population control (not considering any other contributing factors). See Discussion and *Figure 8—figure supplement 3* for more details.

The following figure supplements are available for figure 8:

**Figure supplement 1.** Male pheromone does not affect developmental rates.

**Figure supplement 2.** Male pheromone treatment from egg onward severely affects both lifespan and cross-offspring production.

**Figure supplement 3.** Male pheromone treatment from egg onward reduces offspring and might be a novel mechanism to cull male population in hermaphroditic species.

---

androdioecious species such as *C. elegans*, males are normally rare (0.2%), so the chance that any worm he encounters will be an appropriate mating partner is very high; thus, there may be less selection pressure to evolve pheromones as chemical messengers to identify mates. By contrast, *C. remanei* uses pheromone to distinguish males from females, an important requirement for mating in 50:50 mixed populations. *C.remanei* males are slightly repelled by male pheromone (*Chasnov et al., 2007*) (*Figure 6H*), but are extremely attracted to *C. remanei* female pheromone, while *C. remanei*, as well as both males and females from other gonochoristic species, are immune to male pheromone toxicity (*Figure 6C,D*). Thus, gonochoristic species use pheromones primarily as chemical cues to identify mates, rather than to kill males.

The fact that male pheromone toxicity is present in three distantly-related and separately evolved hermaphroditic *Caenorhabditis* species (*Cho et al., 2004*; *Kiontke et al., 2004; Kiontke et al., 2011*) suggests an important role for male pheromone killing. Periodic explosions of male populations in androdioecious species (e.g., under stressful conditions) allow outcrossing and ensure genetic diversity (*Anderson et al., 2010*). After this beneficial period, however, males are more costly to maintain, and there may be pressure to return to a primarily hermaphroditic population (*Figure 8E*). It is notable that because *C. elegans* males are XO, rather than XY, males may have no selfish drive to maintain their own chromosomes. Using male pheromone as a dose-dependent toxin may be an effective way to cull the male population and ensure that the species returns to the self-reproduction mode when the stressful condition has passed, aiding the return to hermaphroditism. Because a high fraction of males can only be produced by mating (mating produces 50% males, while male production rates from hermaphroditic selfing is 0.2% [*Chasnov and Chow, 2002*; *Hodgkin, 1983*]), the combination of decreased mating efficacy and decreased progeny production might be expected to specifically affect the number of males produced each generation (*Figure 8—figure supplement 3*). Male pheromone alone could effectively drive the population back to a primarily hermaphroditic state after several generations (*Figure 8E*). Previous experiments showed that the average time for males to disappear in N2 strain is 12–20 days (i.e. 4–7 generations) (*Wegewitz et al., 2008*). The discrepancy between our modeling (~15 generations) and the previous experimental result may indicate that multiple factors, including increased hermaphroditic progeny production and decreased mating rates (*Wegewitz et al., 2008*), decreased copulation performance in aging males (*Chatterjee et al., 2013*), and hermaphrodites' response to males [*Garcia et al., 2007*; *Kleemann and Basolo, 2007*; *Morsci et al., 2011*]) could act in tandem with pheromone-dependent killing of males to cull the male population and thus promote a return to hermaphroditism. Male-specific culling occurs in species such as *Drosophila bifasciata,* in which *Wolbachia* infection leads to the killing of male embryos, suggesting that sex ratio can be controlled through male-killing (*Stevens et al., 2001*). Mathematical modeling shows that selection in *C. elegans* favors low

populations of males (*Stewart and Phillips, 2002*), and our model provides a mechanism for how this might be achieved.

In summary, germline-dependent early death after mating is conserved between sexes and perhaps even across great evolutionarily distances, and is likely due to an unavoidable cost of mating, the result of mated animals ramping up germline proliferation and subsequently exhausting their own resources as quickly as possible to produce the next generation of progeny. The differential use of pheromones as toxins or chemical messengers by males in androdioecious and gonochoristic species, respectively, demonstrates that they adopt different strategies to compete, mate, and maintain optimal sex ratios.

## Materials and methods

### Strains
N2 (wild type)
CB4108: *fog-2(q71) V*
CB4037: *glp-1(e2141) III*
DR476: *daf-22(m130) II*
RB711: *pqm-1(ok485) II*
RT130: pwIs23 [vit-2::GFP]
PB4641: *Caenorhabditis remanei*
PB2801*: Caenorhabditis brenneri*
AF16: *Caenorhabditis briggsae*
JU1422: *Caenorhabditis nigoni*
JU1373*: Caenorhabditis tropicalis*
6699 EG4389: *him-5(e1490) V; lin-15(n765ts)X; oxEx860[P(rab-3)::fem-3(wt)::mCherry(worm)::unc-54, pkd-2::gfp(S65C), lin-15(+)]* (gift from the Jorgensen Lab)

### Individual male mating lifespan assays
All lifespan assays were performed at room temperature (~20–21°C), except for *glp-1* male lifespan assays (performed at 25–26°C). 35 mm NGM plates were used for all the experiments in this study. 20 µl of OP50 was dropped onto each plate to make a bacterial lawn of ~10 mm diameter. The next day, one synchronized late L4 male and one late L4 hermaphrodite/female were transferred onto each 35 mm NGM plate. For experiments in *Figures 3C, D* and *4C*, multiple L4 hermaphrodites were transferred together with one male. One late L4 male of the same age and genotype was transferred onto the control plates. Except for *Figure 4A*, *fog-2(q71)* hermaphrodites were used as the *C. elegans* hermaphrodites in the mating assay, because *fog-2* hermaphrodites do not have self sperm, thus allowing us to easily detect successful mating (i.e. eggs and progeny on the plates). In mated groups, **we only included males that were able to produce progeny in our analysis**. However, for the experiments regarding *glp-1* males, mating on FUdR, and the inter-species cross between *C. elegans* males and *C. remanei* females, we included all the males in the analysis. Worms were transferred onto new plates every other day. If the hermaphrodites were lost or bagged, new unmated Day one *fog-2* hermaphrodites were added as replacements. Males and hermaphrodites/females were kept together for 6 days (unless noted otherwise in the text); afterwards only males were transferred on to newly seeded plates every 2–3 days. For RNAi experiments in *Figure 4H*, synchronized eggs were transferred onto NGM plates with RNAi bacteria, late L4 males were transferred and paired with *fog-2* L4 hermaphrodites onto NGM plates seeded with OP50 (to eliminate the possible effect on mating efficiency for different RNAi treatments). Two days later, males and hermaphrodites were transferred onto fresh plates seeded with corresponding RNAi bacteria and males were maintained on RNAi bacteria thereafter. 30–50 worms were included in each group of individual worm lifespan assays. The sample size was similar to previous published study of individual hermaphrodite lifespan assays. When lifespan assays were completed, Kaplan-Meier analysis with log-rank (Mantel-Cox) method was performed to compare the lifespans of different groups. The summary of all lifespan experiments is included in *Supplementary file 1*.

## Grouped males

35 mm NGM plates were used for all the experiments in this study. 20 µl of OP50 was dropped onto each plate to make a bacterial lawn of ~10 mm diameter. The next day, eight synchronized late L4 males were transferred onto each plate. (Two or four males per plate for experiment in *Figure 1A*.) One late L4 male of the same age and genotype was transferred onto the control plates. Males were transferred onto fresh plates every two days, when the males were lost or dead, males from other plates were transferred together to make the size of the group stable.

## Male-conditioned plates (MCP) setup

Male-conditioned plates for lifespan assays were prepared similar to the previous description (*Maures et al., 2014*). Briefly, 60 µl of OP50 was dropped onto each 35 mm NGM plate to make a bacterial lawn of ~25 mm diameter. Young Day 1 wild-type males (*fog-2* males) were transferred onto each plate. Two days later, they were removed and worms for lifespan assays were immediately transferred onto these male-conditioned plates (MCP). These male-conditioned plates were prepared throughout the course of the lifespan assays (*Figure 1—figure supplement 1B*) to ensure fresh MCP plates were available. The number of wild-type males used for conditioning is stated in the text and labeled in the figures. In *Figure 1—figure supplement 1F*, *glp-1* mutant males as well as the wild-type males were used for conditioning at 25°C for 2 days. For MCP treatment from egg onward, 30 Day 1 wild-type males were used to condition plates for 2–3 days. Males were removed and about 15 Day 1 mated *fog-2* hermaphrodites were picked onto these MCP plates for 4–5 hr, allowing them to lay 60–80 eggs. Two days later, L4 males were individually transferred onto MCP plates (conditioned by 8 males, as previously described) for the lifespan assays.

## Body size measurement

Images of live males on 35 mm plates were taken daily for the first week of adulthood with a Nikon SMZ1500 microscope. Image J was used to analyze the body size of the worms. The middle line of each worm was delineated using the segmented line tool and the total length was documented as the body length of the worm. T-test was performed to compare the body size differences between groups of males in the same day. See *Supplementary file 2* for summary.

## FUdR experiment

FUdR was added to the NGM media to the final concentration of 50 µM. Late L4 males and hermaphrodites were transferred onto NGM+FUdR plates seeded with OP50. Worms were transferred every two days, and were kept on FUdR plates for different period of time (3 days, 6 days or lifetime as indicated by text).

## DAPI staining and analysis of male germline

Worms were stained according to Bio-protocol (http://www.bio-protocol.org/wenzhang.aspx?id=77) using VECTASHIELD HardSet Mounting Medium with DAPI from Vector Laboratories (Burlingame, CA). Images were taken with a Nikon Ti. The mitotically proliferating germline region was determined by the crescent shape of DAPI-stained nuclei in the transition zone. Z-series of pictures were taken and the numbers of cells in the mitotically proliferating germline region were counted manually. We scored the germline morphology as '1' (clear transition zone marked by crescent shaped nuclei and sperm after U turn of the germline), '2' (no clear transition zone), and '3' (sperm appear before U turn of the germline). Nonparametric comparison between each treatment group was performed using Prism Graphpad. Mann Whitney test was used to determine the statistical significance.

## Oil Red O staining and quantification

Oil Red O staining was adapted from the published protocol for staining of a small number of worms (*Wählby et al., 2014*). About 20 worms per treatment were imaged with Nikon Ti. Oil Red O quantification was also performed as published (*O'Rourke et al., 2009*). In brief, the color images were split into RGB monochromatic images in Image J. The Oil-Red-O staining arbitrary unit (a.u.) was determined by mean gray value within the worm region by Image J (Intensity in the Green channel was used as the signal, adjusted by the intensity in the Red channel as the background). T-test analysis was performed to compare the fat staining of different groups of worms.

## Glycogen staining

Glycogen staining was performed according to the published protocol (*Frazier and Roth, 2009*). Mating of males was set up as previously described. Right before staining, live males of the same group were picked into an M9 droplet with 1M sodium azide on a 3% agarose pad. Immediately after the liquid was dry, the pad was inverted over the opening of a 50g bottle of iodine crystal chips (Sigma, St. Louis, MO) for 1 min. After the color stained by iodine vapor on the pad disappeared (non-specific staining), the worms (about 20 worms per treatment) were immediately imaged by a Nikon microscope. Due to uncontrollable differences, it is hard to compare the staining performed at different times. Thus, worms from the groups of comparison were mounted onto the same pad (using a separate M9 droplet if there is no visible difference). Image J was used to compare the mean intensity of iodine staining after the background was subtracted. T-test was performed to compare the staining between different groups (on the same pad).

## GFP intensity quantification

10–20 worms of each group were imaged by Nikon Ti. Image J was used to measure the mean and the maximum GFP intensity of the whole body area. T-test analysis was performed to compare the GFP intensity of different groups of worms.

## Mated males microarrays

We paired a single male with a *fog-2* hermaphrodite for about 3.5 days of mating, then picked the males individually on Day four for microarray analysis. As a control, solitary males were collected at the same time. About 150 males (on 150 individual 35 mm plates) were collected for each condition and replicate. Three biological replicates were performed. RNA was extracted by the heat-vortexing method. Two-color Agilent microarrays were used for expression analysis; detailed steps and analysis were performed as we previously reported (*Luo et al., 2010*).

## *daf-22* grouped males microarrays

Synchronized late L4 *daf-22* males were picked on to 35 mm plates (control and MCP). 30 males per plate, 150 males in total were used for each biological replicate. Males were transferred on to freshly seeded plates or MCP plates every two days, and collected on Day six for RNA extraction. Four biological replicates were performed.

## 6699 EG4389 masculinized hermaphrodites grouped vs single microarrays

Synchronized late L4 worms were picked onto 35 mm plates. In the 'grouped' condition, 30 hermaphrodites were picked onto one plate, and ~120 worms were used for each replicate. Worms were transferred every two days to exclude progeny, and were collected on Day six for RNA extraction. Four biological replicates were performed.

## Microarray data accession links

Microarray data can be found in PUMAdb (http://puma.princeton.edu).
 https://puma.princeton.edu/cgi-bin/publication/viewPublication.pl?pub_no=576
Including mated males microarrays (three biological replicates);
*daf-22* grouped males microarrays (four biological replicates);
*glp-1* hermaphrodites treated with MCP microarrays (four biological replicates);
EG4389 masculinized hermaphrodites grouped vs single microarrays (four biological replicates).

## Analyses of microarray data

Significant differentially-expressed gene sets were identified using SAM (*Tusher et al., 2001*). Previously reported microarray results exploring the effect of males on hermaphrodites (*Maures et al., 2014*) were downloaded from NCBI and compared to our differentially expressed gene lists. Enriched motifs were found using RSAT (*van Helden, 2003*).

## Pheromone chemotaxis assay

This assay (*Figure 6H*) was modified from a previous assay (*Chasnov et al., 2007*). 10 Day 1 virgin *C. remanei* or *C. elegans* hermaphrodites were placed in 100 µl of M9 buffer at room temperature overnight with shaking. 100 males of either *C. elegans* or *C. remanei* were placed in 100 µl of M9. The supernatant solutions were then used for the pheromone chemotaxis assay. 60 mm NGM plates (no food) were used for the chemotaxis assay. Two destination spots (supernatant and M9 control) were separated by about 45 mm; the distance from the origin spot to either destination spot is 30 mm. Two 1 µl drops of 1M sodium azide were first applied to the destination spots. When dry, a drop of 1 µl M9 or supernatant was separately added onto the destination spots. Then, over 10 young adult (Day 2) males were placed at the origin spot, transferring as little bacteria as possible. After 60 min, the paralyzed male worms were scored based on their location. The chemotaxis index was calculated as: (#worms at supernatant destination - #worms at control destination)/(#total worms - #worms at origin). The chemotaxis assay in *Figure 2B* was also modified from established protocol (*Chasnov et al., 2007*). 10 Day 5 hermaphrodites of either N2 or 6699 EG4389 were put in 100 µl of M9 buffer at room temperature overnight with shaking. Two destination spots were 3 mm apart. The origin spot was in the middle. 20 Day 1 *fog-2* males were used in each assay; two replicates were performed.

## Male fertility assay

Males from the control plates and MCP plates were individually paired with one virgin Day 1 *fog-2* hermaprhodite at various time points on a seeded 35 mm NGM plate for 24 hr. About 20 pairs were set up for each group in each biological replicate. The percent fertile was calculated from the number of plates with eggs/progeny divided by the total number of plates set up for this group. Each mated hermaphrodite was numbered and was transferred individually onto a new seeded NGM plate every day to count the total progeny/male number.

## Acknowledgements

We thank the Caenorhabditis Genetics Center (CGC) and E Jorgensen for strains, Z Gitai and N Wingreen for valuable discussions, and members of the Murphy laboratory for critically reading the manuscript. CS is supported by March of Dimes, and AMR by NIH 5T32GM007388-39. CTM is supported by NIH DP1 GM119167-02, HHMI Faculty Scholars Award, and is the Director of the Glenn Center for Aging Research at Princeton.

## Additional information

### Funding

| Funder | Grant reference number | Author |
|---|---|---|
| Glenn Foundation for Medical Research | | Cheng Shi<br>Coleen T Murphy |
| National Institutes of Health | DP1 GM119167-02 | Cheng Shi<br>Alexi M Runnels<br>Coleen T Murphy |
| National Institutes of Health | 5T32GM007388-39 | Cheng Shi<br>Alexi M Runnels |
| Howard Hughes Medical Institute | Faculty Scholars Award | Coleen T Murphy |

The funders had no role in study design, data collection and interpretation, or the decision to submit the work for publication.

### Author contributions

CS, AMR, Conceptualization, Data curation, Software, Formal analysis, Validation, Investigation, Visualization, Methodology, Writing—original draft, Writing—review and editing; CTM,

Conceptualization, Resources, Supervision, Funding acquisition, Writing—original draft, Project administration, Writing—review and editing

**Author ORCIDs**
Cheng Shi, http://orcid.org/0000-0003-0365-8273
Alexi M Runnels, http://orcid.org/0000-0002-8592-1444
Coleen T Murphy, http://orcid.org/0000-0002-8257-984X

## Additional files

**Supplementary files**

• Supplementary file 1. Summary of all lifespan assays performed in this study.

• Supplementary file 2. Summary of body size measurements.

• Supplementary file 3. Significantly up- and down-regulated genes in mated males identified by Significance Analysis of Microarrays (FDR = 0%).

• Supplementary file 4. Significantly up- and down-regulated genes under male pheromone-induced conditions identified by Significance Analysis of Microarrays (FDR = 1%). *Sheet 1*: grouped vs. solitary neuronally-masculinized hermaphrodites. *Sheet 2*: grouped *daf-22* males on plates conditioned by wild-type males vs. grouped *daf-22* males on control plates. *Sheet 3*: overlap between the two conditions.

• Supplementary file 5. Comparison of the list of genes significantly up- and down-regulated upon MCP treatment of *daf-22* males to previously published arrays of male pheromone-treated hermaphrodites (*Maures et al., 2014*). *Sheet 1*: comparison of up-regulated genes. *Sheet 2*: comparison of down-regulated genes.

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
