## [Decision Letter]

Thank you for submitting your article "Mating and Male pheromone kill *Caenorhabditis* males through distinct mechanisms" for consideration by *eLife*. Your article has been favorably evaluated by Marianne Bronner (Senior Editor) and three reviewers, one of whom, Andrew Dillin (Reviewer #1), is a member of our Board of Reviewing Editors. The following individuals involved in review of your submission have agreed to reveal their identity: L René García (Reviewer #2); Maureen M Barr (Reviewer #3).

The reviewers have discussed the reviews with one another and the Reviewing Editor has drafted this decision to help you prepare a revised submission.

The article "Mating and male pheromone kill *Caenorhabditis* males through distinct mechanisms" explores how male secretions and copulation shortens a *C. elegans* male's lifespan relative to a male that is reared in isolation. We find the research topic very interesting and especially novel, since so very few *C. elegans* labs study male *C. elegans* development, behavior and physiology. The authors incorporate life span quantification along with gene profiling analyses, which they have used extensively in their past studies involving hermaphrodites, to describe anatomical and physiological changes that occur in aging adult *Caenorhabditis* sp. Males. The authors investigate life history traits in males that are chronically exposed to male secretions (which includes but is not limited to ascaroside blends) and to hermaphrodite/female mates.

While all reviewers are in favor of publication of this body of work, we all think that the discussion and use of the term "toxin" to describe male pheromone needs to be tempered and explained better.

In the last paragraph of the Introduction, the authors immediately introduce the male secretions as a toxin. Throughout the article, the authors convincingly show that chronic exposure to male conditioned media has toxic properties, but immediately calling the male secretions a toxin requires the concept to be developed further. The connotation of a "toxin" is that it has a deterministic role to induce deleterious physiological effects that ultimately aids in the fitness of the organism. The experiments presented in this paper do not directly address any organismal fitness benefits for the conditioned media's toxic properties. Although in the Discussion (subsection “Male pheromone-induced killing may be a strategy to selectively reduce the male population”), the authors speculate that the N2 male secretions would cull males from the population, details of how this scenario would occur needs to be explained further.

Finally, the authors should also incorporate into their Discussion how copulatory behaviors rapidly decline in early adulthood and how this could affect reproduction of males in their prime.

Male copulation performance declines between day 2 to day 5, and this behavioral decline does not change whether the male is reared in isolation or in large groups (i.e. the male secretions has no obvious deleterious effect on the young male's copulatory motor execution or fecundity). Thus, if male secretions are made to cull out males, then its role is to cull out males past their reproductive prime, but this scenario should not change the generation of new males from cross-fertilization, if male pheromone started killing off young virile males. The reduction of new males in hermaphroditic population over time is more likely due to inefficient male copulatory motor execution, relative to hermaphroditic selfing.

---

## [Author Response]

[…] While all reviewers are in favor of publication of this body of work, we all think that the discussion and use of the term "toxin" to describe male pheromone needs to be tempered and explained better.

In the last paragraph of the Introduction, the authors immediately introduce the male secretions as a toxin. Throughout the article, the authors convincingly show that chronic exposure to male conditioned media has toxic properties, but immediately calling the male secretions a toxin requires the concept to be developed further. The connotation of a "toxin" is that it has a deterministic role to induce deleterious physiological effects that ultimately aids in the fitness of the organism. The experiments presented in this paper do not directly address any organismal fitness benefits for the conditioned media's toxic properties. Although in the Discussion (subsection “Male pheromone-induced killing may be a strategy to selectively reduce the male population”), the authors speculate that the N2 male secretions would cull males from the population, details of how this scenario would occur needs to be explained further.

We have now tried to explain this point better in the Introduction, which summarizes our results: “that is, while male pheromone itself is not a general poison to all worms, its perception by *C. elegans* males leads to death and to male-specific reproductive decline”.

One of the important findings here is that different sexes and species perceive male pheromone differently; *C. remanei* uses male pheromone primarily as an attraction/repelling mating cue, while *C. elegans* males are exquisitely sensitive to and die from male pheromone, and *C. elegans* hermaphrodites are less susceptible to this killing; because the male pheromone kills the worms, we have used the term “toxin”, but we have eliminated it in several places (and we are open to other suggestions).

Finally, the authors should also incorporate into their Discussion how copulatory behaviors rapidly decline in early adulthood and how this could affect reproduction of males in their prime.

Male copulation performance declines between day 2 to day 5, and this behavioral decline does not change whether the male is reared in isolation or in large groups (i.e. the male secretions has no obvious deleterious effect on the young male's copulatory motor execution or fecundity). Thus, if male secretions are made to cull out males, then its role is to cull out males past their reproductive prime, but this scenario should not change the generation of new males from cross-fertilization, if male pheromone started killing off young virile males. The reduction of new males in hermaphroditic population over time is more likely due to inefficient male copulatory motor execution, relative to hermaphroditic selfing.

We agree with this logic, and frankly, it was one of the points that had bothered us throughout this work: why would this male-specific killing be conserved across different androdioecious species if there is no observable fitness effect, which must act prior to the end of reproduction?

We have performed new experiments to directly address these points, and we are excited to include them in the revision. Our previous experiments were all aimed at testing adult lifespan effects, and in order to distinguish those from developmental effects, the application of worms to male pheromone plates was performed as adults. Therefore, our results would not include any effects that would be caused by egg or larval exposure to male pheromone. We have now found that when eggs are placed onto male pheromone, the adult males are less fertile, with fewer successful matings, and lower brood sizes from those matings, in addition to an even shorter lifespan. (We confirmed that there was an age-related decrease in male fertility, as previously reported (Chatterjee et al. 2013; Guo, et al. 2012), and under our conditions, male pheromone seems to have exacerbated this effect.) Together, the impact of decreased male fertility and decreased brood sizes, while hermaphrodite brood size was unaffected by MCP, might reduce the male fraction of the population over several generations. We have added a simple model of this effect to the Discussion.